# Achieving three-dimensional lithium sulfide growth in lithium-sulfur batteries using high-donor-number anions

Hyunwon Chu[1], Hyungjun Noh[1], Yun-Jung Kim[1], Seongmin Yuk[1], Ju-Hyuk Lee[1], Jinhong Lee[1], Hobeom Kwack[1], YunKyoung Kim[2], Doo-Kyung Yang[2] & Hee-Tak Kim [1,3]

Uncontrolled growth of insulating lithium sulfide leads to passivation of sulfur cathodes, which limits high sulfur utilization in lithium-sulfur batteries. Sulfur utilization can be augmented in electrolytes based on solvents with high Gutmann Donor Number; however, violent lithium metal corrosion is a drawback. Here we report that particulate lithium sulfide growth can be achieved using a salt anion with a high donor number, such as bromide or triflate. The use of bromide leads to ~95 % sulfur utilization by suppressing electrode passivation. More importantly, the electrolytes with high-donor-number salt anions are notably compatible with lithium metal electrodes. The approach enables a high sulfur-loaded cell with areal capacity higher than 4 mA h cm$^{-2}$ and high sulfur utilization ( > 90 %). This work offers a simple but practical strategy to modulate lithium sulfide growth, while conserving stability for high-performance lithium-sulfur batteries.

[1] Korea Advanced Institute of Science and Technology, 291 Daehak-ro, Yuseong-gu, Daejeon 34141, Republic of Korea. [2] LG Chem., LG Science park E6 Block, 30 Magokjungang 10-ro, Gangseo-gu, Seoul 07796, Republic of Korea. [3] Advanced Battery Center, KAIST Institute for the NanoCentury, Korea Advanced Institute of Science and Technology, 335 Gwahangno, Yuseong-gu, Daejeon 34141, Republic of Korea. These authors have contributed equally: Hyunwon Chu, Hyungjun Noh.  Correspondence and requests for materials should be addressed to H.-T.K. (email: heetak.kim@kaist.ac.kr)

With increasing demand for energy-dense batteries that can overcome limits of conventional lithium (Li) ion technology, lithium–sulfur (Li–S) batteries are considered the most promising candidate because of their high energy density (2600 W h kg$^{-1}$) and structural similarity with Li-ion batteries[1,2]. However, practical applications of Li–S batteries are still hampered by intrinsic problems such as the low conductivity of sulfur and lithium sulfide (Li$_2$S)[3–5], large volumetric changes of the electrode[6–8], and dissolution of intermediate lithium polysulfide (LiPS) species during cycling[9–11]. These limitations result in low sulfur utilization[12,13], low coulombic efficiency[14,15], and fast capacity fading[16,17] of Li–S batteries. In particular, during discharge, uncontrolled electrodeposition of Li$_2$S, the final discharge product, is a critical obstacle to achieving complete sulfur utilization. Due to its electronic and ionic insulating property, a film of Li$_2$S covering the cathode causes early electrode passivation and consequent large polarization. This impedes discharge before complete use of the loaded sulfur[12,18]. The problem becomes more significant at higher current density and higher areal sulfur loading, hindering rational design of practical Li–S batteries (Supplementary Fig. 1).

One approach for suppressing electrode passivation is uniform distribution of sulfur species, either by confining them physically in structured host materials such as porous carbons[19,20], graphene materials[21,22], and conductive polymers[23,24] or by chemically adsorbing them onto structures that include transition metal chalcogenides[25–27] and metal organic frameworks (MOFs)[28,29]. These approaches effectively enhance sulfur utilization and suppress the polysulfide (PS) shuttle; however, the electrode passivation issue is not completely resolved during repeated cycles. Another option would be employing a three-dimensional (3D) conducting network with macro channels[30,31]. Macro-pore structures relieve the clogging of ion transport channels by Li$_2$S deposits. Nevertheless, the discharge capacity remains limited due to passivation of the conducting network.

The approaches discussed above feature control of either the cathode material or the structure; conversely, modulating the intrinsic chemistry of Li$_2$S growth by electrolyte optimization could offer a fundamental solution for achieving high sulfur utilization. It has been reported that solvents with high Gutmann Donor Number (DN) promoted the redox reaction of sulfur species by stabilizing multiple states of PS anions and altering the kinetic pathway[32]. By using a high-DN solvent, dimethylacetamide (DMA), full utilization of sulfur and Li$_2$S was achieved based on enhanced chemical transformation between Li$_2$S and soluble PS anions[33]. More recently, variation of Li$_2$S deposition morphology was reported in response to the solvent DNs. Li$_2$S films were observed with the low-DN solvent, whereas flower-like Li$_2$S particles grew with the intermediate and high-DN solvents[34]. The amended Li$_2$S morphologies are in line with the electrolyte dependency of lithium peroxide (Li$_2$O$_2$) morphology in Li–air batteries; electrolytes with high electron donating ability augment the solubility of lithium superoxide (LiO$_2$) and produce 3D toroidal-shaped Li$_2$O$_2$[35]. Although high-DN solvents offer advantages, they have the formidable drawback of extreme reactivity with a Li metal electrode. For this reason, stable cycling of Li–S cells has not been achieved with high-DN solvent-based electrolytes without advanced Li metal protection. Thus, practical application of high-DN solvents for Li–S batteries remains challenging[33,34].

Herein, we suggest a supporting salt anion as an agent to control the electron-donating property of an electrolyte. By changing a salt anion into one with a higher-DN, we induced 3D particulate growth of Li$_2$S. Li$_2$S growth behaviors were examined critically for three salt anions with different DNs in a conventional 1,3-dioxolane (DOL):1,2-dimethoxyethane (DME) (1:1)

solvent. The key advantage of this salt-modification approach is that Li$_2$S growth chemistry can be modulated without severe deterioration of Li metal electrodes, which is unattainable with high-DN solvent electrolytes. With minimized electrode passivation, high sulfur utilization (~95%) and stable cycling were achieved using an extremely small surface area cathode (0.9908 m$^2$ g$^{-1}$), even without additional Li metal protection. In addition, the investigation on the deposition mechanisms with the different salt anions revealed that the high-DN anions affect the partial solubility of Li$_2$S and then trigger 3D growth of Li$_2$S.

## Results

**Discharge and charge behaviors with high-DN anions.** To prove the salt anion effect on the Li$_2$S deposition mode, bistriflimide ((CF$_3$SO$_2$)$_2$N$^-$, TFSI$^-$), triflate (CF$_3$SO$_3^-$, Tf$^-$), and bromide (Br$^-$) anions (of which the DNs are 5.4, 16.9, and 33.7 kcal mol$^{-1}$, respectively[36–38]) were selected and compared. For easier detection of the Li$_2$S morphology, a carbon paper (CP) electrode with a small Brunauer–Emmett–Teller (BET) surface area of 0.9904 m$^2$ g$^{-1}$ (Supplementary Fig. 2) was employed. Fig. 1 shows the electrochemical performances of the Li–S cells with LiPS electrolytes containing the three different salt anions. The theoretical areal capacities of the test cells were set to 1.68 mA h cm$^{-2}$. As shown in Fig. 1a, the conventional lithium bistriflimide (LiTFSI) electrolyte with the lowest electron donating ability delivers only a low capacity of ~400 mA h g$^{-1}$, corresponding to 25% of sulfur utilization due to absence of the lower discharge plateau. The capacity of discharge plateau between 2.2 V and 1.8 V is induced by a reduction of short-chain LiPS (Li$_2$S$_x$, $x \le 4$) to Li$_2$S and precipitation of Li$_2$S. Thus, the sudden voltage drop at the beginning of the lower plateau indicates rapid electrode passivation by Li$_2$S deposition, which limits a further reduction of the LiPS remaining in the electrolyte. In sharp contrast, the lithium triflate (LiTf) and lithium bromide (LiBr) electrolytes show significant extensions of the lower plateaus and result in high discharge capacities of 1214 mA h g$^{-1}$ and 1535 mA h g$^{-1}$, which are 73% and 92% of the theoretical capacity, respectively. Because the lower plateau reaction is mainly limited by electrode passivation from insulation by Li$_2$S, extension of the lower voltage plateau with increasing the DN of the anion suggests that the high-DN anions can retard the surface passivation. For the LiBr electrolyte, the capacity ratio between the upper and lower discharge plateaus (387 mA h g$^{-1}$ and 1148 mA h g$^{-1}$) was 1:3, the same as the theoretical value. This ratio supports the notion that all the short-chain LiPS species generated from the upper plateau reaction were completely converted to Li$_2$S at the end of discharge. More importantly, the charge polarization as well as discharge polarization decreased when the DN of the anion increases, reflecting not only that the electrode passivation can be lowered, but that Li$_2$S decomposition can also be accelerated under the high-DN anion environment. The observed behaviors are quite similar to those reported in studies of high-DN solvents[33,34]. Because of the enhanced PS solubility in high-DN solvents, the chemical decomposition of octa-sulfur (S$_8$) or Li$_2$S is promoted, then the polarization consequently decreases. It appears that the high-DN salt anions function similarly to high-DN solvents during electrochemical operations. However, the high-DN salt anions have an advantage over the high-DN solvents in that the anions provide stable cycling without the additional Li metal protection by inorganic conductors or highly concentrated electrolytes[33,34]. As shown in Fig. 1b, the LiTf and LiBr electrolytes exhibit stable behaviors at 0.2 C over 80 cycles, maintaining their initial capacities. In addition, the coulombic efficiencies of the LiTf and LiBr electrolytes were even higher (>98%) than the efficiency of the conventional LiTFSI electrolyte (96.5 %) (Fig. 1c). These results indicate that the LiTf

and LiBr electrolytes have a higher degree of reversibility for $Li_2S$ formation and decomposition reactions than does the LiTFSI electrolyte. Moreover, even at a higher current density of 0.5 C, the electrolytes with the high-DN salt anions maintained their role in enhancing the discharge capacities (LiTf and LiBr, 994 mA h g$^{-1}$ and 1310 mA h g$^{-1}$, respectively) and enabled reasonably stable cycling (Supplementary Fig. 3).

**Lithium sulfide morphology and electrode passivation.** For the three electrolytes, the evolution of $Li_2S$ morphology was investigated using scanning electron microscopy (SEM) to verify the origin of the extended lower plateau. The SEM images of the cathodes taken at different states of discharge (SOD) are presented in Fig. 2. At the specified capacity stages marked as "1", "2", and "3" along the discharge (Fig. 2a), the cells were individually disassembled and imaged. Comparison of the cathode with the LiTFSI electrolyte (Fig. 2b, c) and the pristine electrode (Fig. 2d) confirms that the carbon surfaces are mostly covered by the laterally deposited $Li_2S$. The insulating $Li_2S$ layer blocks the electrochemical reduction of short-chain LiPS to $Li_2S$, thereby causing the earlier failure of the lower plateau reaction. However, with the higher-DN anions (LiTf and LiBr), 3D growth of $Li_2S$ was induced, creating particle-like $Li_2S$ deposits. As shown in Fig. 2e–g and h–j for the LiTf and LiBr electrolytes, respectively, globular nuclei formed at the beginning of the lower plateau and then grew in three dimensions. By the end of the discharge, the $Li_2S$ particles had evolved to several micrometer-sized deposits on the fiber surfaces and in the interstitial spaces. At the same discharge capacity of ~400 mA h g$^{-1}$ (Stage 2), the carbon surfaces were acutely passivated when using the LiTFSI electrolyte, whereas those for the LiTf and LiBr electrolytes were nearly uncovered because of the 3D $Li_2S$ growth. These results clearly demonstrate that 3D $Li_2S$ growth, induced by the high-DN salt electrolytes, delayed surface passivation of the electrode and enabled high sulfur utilization. Gerber and co-workers previously reported that 3D $Li_2S$ growth induced by a redox mediator suppressed the surface passivation and augmented the lower plateau capacity in the system[39]. The same phenomenon was realized simply by modifying the property of the salt anions, and the morphological discrepancies observed in the SEM analysis agree with the enhanced cell performances of the high-DN anions in Fig. 1. Moreover, the $Li_2S$ decomposition behaviors during the subsequent charge step were also examined using SEM analysis (Supplementary Fig. 4). For Li–S systems, not only capacity enhancement, but also the cycle reversibility must be attained. The images of the cathodes in mid-charge, marked C1, suggest that $Li_2S$ decomposition was initiated from the deposits on the fiber networks, and was then transferred to bulk particles in the interspaces due to the active electron transfer through the carbon fiber networks. At the end of charge (Stage C2), the carbon surfaces of all three electrode samples were completely recovered. The observation proves that regardless of the morphological differences of the deposited $Li_2S$, reversible cycling can be achieved without severe active mass losses even for the high-DN electrolytes.

To understand the influence of the $Li_2S$ morphology on electrode polarization, electrochemical impedance spectroscopy (EIS) analysis of the cathodes was conducted with a three-electrode pouch-type cell configuration (inset of Fig. 2 and Supplementary Figs. 5, 6). For the LiTFSI, LiTf, and LiBr electrolytes, semi-circles in the frequency range of 1–10 Hz, which are mainly associated with the charge transfer process[40,41], expanded as the discharge reaction proceeded (Supplementary Fig. 6). The increase of the semi-circle through the discharge generally reflects that $Li_2S$ electrodeposition elevates the charge

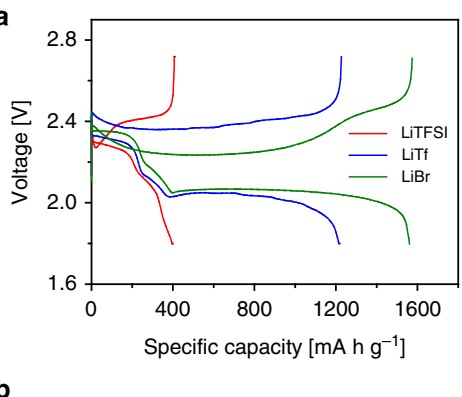

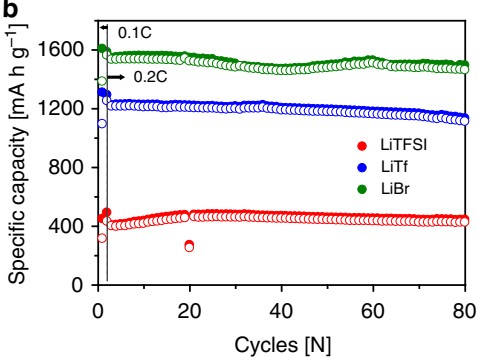

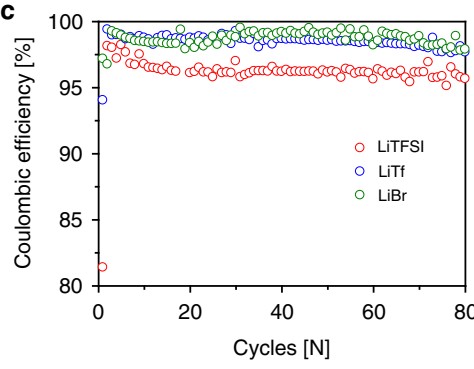

**Fig. 1** Electrochemical performances of the lithium–sulfur cells with varying anions. **a** Charge and discharge curves of the first 0.2 C cycle, **b** comparison of the charge (closed circle) and discharge (open circle) capacities, and **c** Coulombic efficiency data for 80 charge/discharge cycles at 0.2 C. The electrolytes consist of 0.2 M lithium polysulfide (LiPS, $Li_2S_8$ based) with 1 M Li salts LiX, X = bistriflimide (TFSI$^-$), triflate (Tf$^-$), or bromide (Br$^-$) / 0.2 M lithium nitrate (LiNO$_3$) / 1,3-dioxolane (DOL):1,2-dimethoxyethane (DME) (1:1)

transfer resistance ($R_{ct}$) due to the insulating nature of $Li_2S$. However, the evolution of $R_{ct}$ values was different among the salt anions. Comparison of the impedances at the beginning and the shallow discharge of the lower plateau (Stage 1, 2) shows that the increase of $R_{ct}$ values was much smaller for the high-DN salt anions than for the low-DN anion. This verifies that electrode passivation can be delayed with the high-DN anions by the 3D growth of $Li_2S$. More importantly, at the end of the discharge, the impedances for the LiTf and LiBr (Stage 3) electrolytes displayed capacitive behaviors that feature drastic increase of the imaginary value ($-Z''$) without the appearance of a semi-circle. The observed capacitive behaviors suggest that for the LiTf and LiBr, the lower plateau reaction was limited not by electrode passivation but by depletion of the active materials nearby; because all LiPS molecules in contact with the conducting surface were consumed, the charge transfer process did not appear in the impedance

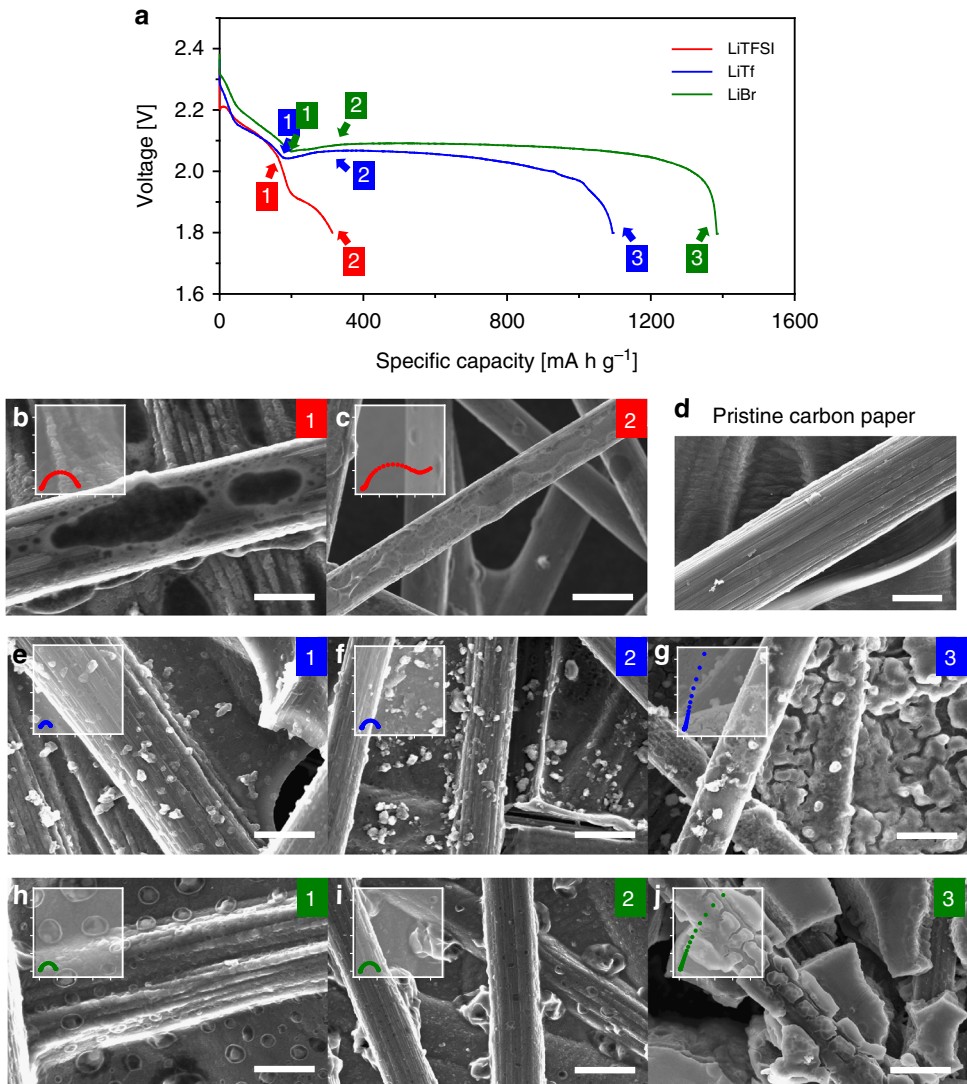

**Fig. 2** Morphological evolutions of lithium sulfide at different states of discharge. **a** The first discharge curves at 0.05 C. Discharge of the individual cells stopped at the marked states of discharge (SOD) for Scanning electron microscopy (SEM) analysis. **d** A SEM image of the pristine carbon paper (CP) electrode, SEM images of the CP electrodes at different SODs with the **b**, **c** lithium bistriflimide (LiTFSI), **e–g** lithium triflate (LiTf), and **h–j** lithium bromide (LiBr) electrolytes. Scale bars, 5 μm (**b**, **d**, **e**, **h**); 10 μm (**c**, **f**, **g**, **i**, **j**). The Nyquist plots of the impedances measured at the points are provided in the insets. (The range of the x and y-axes is from −100 to 1400 Ω cm$^2$; enlarged figures are given in Supplementary Fig. 6.)

spectroscopy. In addition, as shown in Supplementary Fig. 6, after-charge impedances of the cathodes recovered to the same values as those before discharge, despite the formation of micron-sized Li$_2$S deposits under the high-DN salt conditions. This is in accord with the reversible decomposition of Li$_2$S deposits observed by SEM analysis in Supplementary Fig. 4.

The delayed surface passivation when discharging with high-DN salt anions is noteworthy. Because the excessive carbon surface is no longer needed for the complete conversion of the active material, this approach allows the paradigm shift in sulfur cathode design from expanding the electrode surface that accommodates 2D grown Li$_2$S, to creating specific void spaces to uptake 3D grown Li$_2$S. Although the SEM analysis revealed the 3D Li$_2$S morphology when employing the high-DN anions, the effect on the evolution of carbon surfaces during the discharge was not completely ruled out. For an in-depth characterization of the electrodes, X-ray photoelectron spectroscopy (XPS) was conducted for the CP electrodes after discharging the same

capacity. As shown in the survey scan of the discharged cathodes (Fig. 3a), the intensity of the C 1s peak of the electrode with LiTFSI is much lower than the peaks with the LiTf and LiBr electrolytes. The intensity difference came from a huge diminution of the C–C bond (284.7 eV)[42–44] for the LiTFSI sample (Fig. 3b). Simultaneously, as Fig. 3c presents, the S 2p peak from Li$_2$S at 160.2 eV[45,46] appears much larger with the LiTFSI electrolyte than with the others. Thus, these XPS results complement the previous observation that the carbon surfaces discharged with the LiTFSI electrolyte were rapidly passivated due to the surface covering by Li$_2$S. On the other hand, by virtue of the 3D Li$_2$S growth, most carbon surfaces remained intact with the LiTf and LiBr electrolytes even at the same discharge state. Thus, the 3D growth enabled the full conversion of the discharge intermediates, leading to high sulfur utilization. Additionally, the different carbon passivation tendency according to the anion characteristic was independently verified once more by X–ray diffraction (XRD) analysis; the peaks at 27° and 55° associated

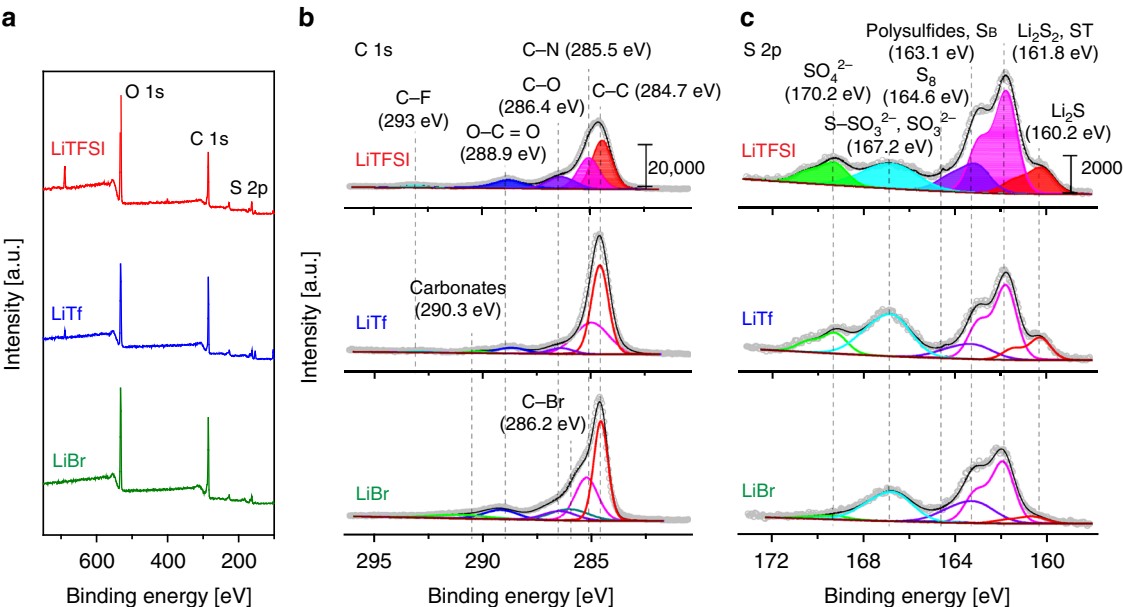

**Fig. 3** Surface characterization of discharged carbon paper electrodes. **a** X-ray photoelectron spectroscopy (XPS) survey scans of the carbon paper (CP) electrodes after discharging 400 mA h g⁻¹ with the catholytes using the three different salts, lithium bistriflimide (LiTFSI), lithium triflate (LiTf), and lithium brom0ide (LiBr). **b** Comparisons of C 1s, and **c** S 2p spectra of the discharged cathodes with the three electrolytes

with crystalline carbon were diminished only with the electrode sample of the LiTFSI electrolyte (Supplementary Fig. 7).

The dynamic growth trajectories of Li₂S deposition, deviated by the salt anions, were studied using chronoamperometry (CA). The CA technique is unique, enabling in situ characterization on the kinetics and morphology of an electrodeposited material, according to a current response from the potentiostatic driving force. Before the analysis, all cells were potentiostatically pre-discharged to 2.2 V to eliminate long-chain LiPSs ($Li_2S_x$, $x > 4$) and to extract the current response solely from Li₂S electro-deposition for the CA analysis. The deposition did not take place above 2.2 V as verified by the cyclic voltammetry (CV) data (Supplementary Fig. 8), and by the current responses at different potentiostatic discharges (Supplementary Fig. 9). After pre-discharge, the voltage was stepped down to 2.0 V and maintained to initiate Li₂S deposition. The initial current decreased exponentially because of a reduction of the remaining short-chain LiPSs. The current then rose as the Li₂S deposition progressed. After the reaction current reached the maximum value, it decayed due to impingement of the growing Li₂S deposits and passivation of the surface by insulating Li₂S[47,48]. Therefore, the $I_m$ and $t_m$, which are the maximum current and its corresponding time in the chronoamperogram, can be used to interpret the relative speed of carbon surface passivation under a different electrolyte system. (i.e. one that shows the lower $t_m$ has the faster passivation speed.) The applied voltage of 2.0 V is in the "kinetic-controlled" regime because the voltage is close to or higher than the second reduction peak potentials in Supplementary Fig. 8. The monitored current responses were construed using the Bewick, Fleischmann, and Thirsk (BFT) instantaneous theory model[49,50] (Supplementary Fig. 10 and Supplementary Table 1). Significantly, assuming other parameters in the equation are constant among the three electrolytes, the term $N_0 \, k_g^2$ represents the lateral growth rate constant of Li₂S, and thus can be used as a measure of the relative electrode passivation speed[47,48]. The $N_0 \, k_g^2$ values for the three electrolytes were calculated based on the $t_m$ from Fig. 4a, and compared in Fig. 4b.

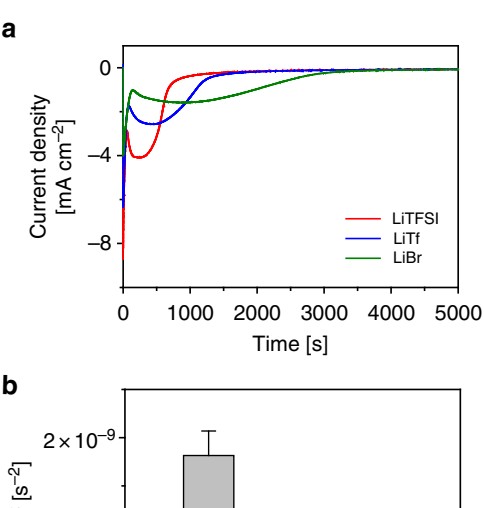

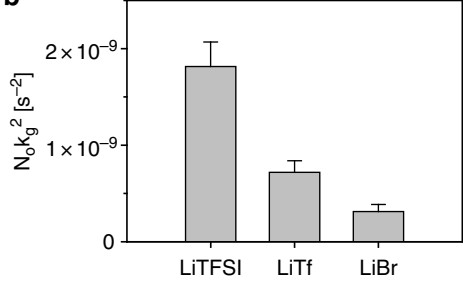

**Fig. 4** Chronoamperometry analysis on electrodeposition behaviors of lithium sulfide. **a** Current vs. time curves for potentiostatic electrodepositions of 1 M LiX, X = bistriflimide (TFSI⁻), triflate (Tf⁻), or bromide (Br⁻), and 0.2 M lithium polysulfide (LiPS) catholytes using carbon nanotube (CNT) electrodes. **b** The lateral growth rate constants ($N_0 \, k_g^2$) of lithium sulfide (Li₂S) depositions with different electrolytes are calculated based on the Bewick, Fleischmann, and Thirsk (BFT) instantaneous model. The provided data are the average values from the five independent chronoamperometry (CA) experiments. The error bars indicate the standard errors. Source data are provided as a Source data file

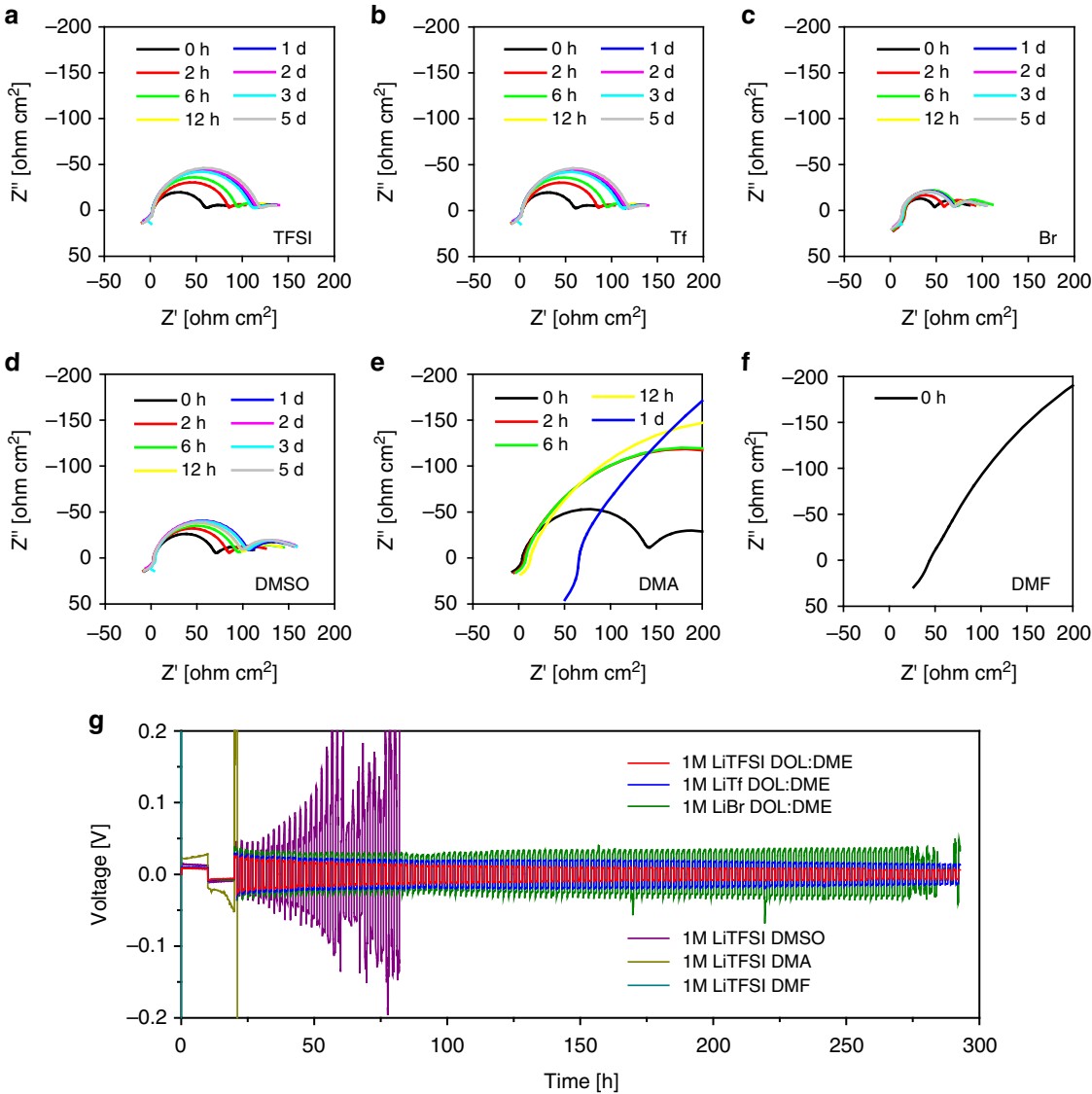

**Fig. 5** The stability of lithium metal electrodes with different electrolyte combinations. To verify the stability of lithium (Li) metal electrodes with different salt/solvent compositions, Li/Li symmetric cells were assembled employing six salt/solvent modified electrolytes. **a–f** Changes in impedances of the symmetric cells were observed under the five-day storage condition, **g** Galvanostatic 1 h/1 h charge/discharge operation was conducted with 0.5 mA cm$^{-2}$ current density for six different electrolyte systems. The six salt/solvent modified electrolytes include 1 M LiX, X = bistriflimide (TFSI$^-$), triflate (Tf$^-$), or bromide (Br$^-$), in 1,3-dioxolane (DOL):1,2-dimethoxyethane (DME) (1:1), and 1 M LiTFSI in dimetylsulfoxide (DMSO), dimethylacetamide (DMA), and dimethylformamide (DMF)

As a result, the LiBr electrolyte showed the lowest lateral growth rate of $3.30 \times 10^{-10}$ s$^{-2}$, which is 2.5 times slower than that of the LiTf electrolyte ($7.45 \times 10^{-10}$ s$^{-2}$) and 5.5 times slower than that of the LiTFSI electrolyte ($1.82 \times 10^{-9}$ s$^{-2}$). The outcome stands that the electrode with the LiBr electrolyte was passivated 5.5 times slower than that with the LiTFSI electrolyte, which matches with the extension of the lower plateau in Fig. 1. The sets of ex situ examinations and in situ electrochemical measurements shared one common understanding: 3D growth of Li$_2$S, engendered by the high-DN salt anions, effectively suppressed the surface passivation and finally led to an increase in the cell capacity.

**Compatibility with lithium metal electrodes**. The high-DN solvents such as dimethylsulfoxide (DMSO), dimethylaceta-mide (DMA), and dimethylformamide (DMF) can enhance sulfur utilization, as demonstrated in previous research.

However, these solvents intensely corrode Li metal electrodes, hindering their application for Li–S batteries. Therefore, Li metal stability with high-DN salt anions is of great importance. In this regard, the storage stability and cycling stability of Li metal electrodes were assessed for six salt/solvent varied electrolytes: DOL:DME-based 1 M LiTFSI, LiTf, and LiBr electrolytes, and 1 M LiTFSI electrolytes with high-DN solvents (DMSO, DMA, and DMF). As shown in Fig. 5a–f, the DMA- and DMF-based electrolytes showed dramatic increases of their interfacial resistances, indicating severe Li metal corrosion. However, the interfacial resistance for the DMSO-based electrolyte became invariant with time. The DOL:DME-based LiTFSI, LiTf, and LiBr electrolytes also exhibited eventual stabilization of the interfacial resistances. These results suggest that the DMSO and DOL:DME-based electrolytes can form enough solid electrolyte interphase (SEI) layers that prevent continuous Li metal corrosion under the idle condition. Next, the cycling stabilities of the Li/Li symmetric cells with the six

different electrolytes were compared in Fig. 5g. For the DMA and DMF electrolytes, the operations of the cells failed immediately upon the first formation cycle of $0.1\,mA\,cm^{-2}$. The DMSO-based electrolyte showed longer cycling ability at 0.5 $mA\,cm^{-2}$, but the overvoltage of the DMSO cell began to fluctuate only within 20 plating/stripping cycles. In definite contrast to the high-DN solvent systems, the DOL:DME-based electrolytes provided stable operation at $0.5\,mA\,cm^{-2}$ for more than 250 plating/stripping cycles. Therefore, the salt modification strategy, compared to the solvent control, would serve as a more effective approach for Li–S full cell design from the viewpoint of maintaining compatibility with a Li metal electrode.

The enhanced Li metal stability of the salt-modified electrolytes can be explained by change in the solvation cluster of Li ion ($Li^+$). In a non-aqueous electrolyte, solvents and anions are the two major chemical components that participate in a $Li^+$ solvation cluster[51]. The whole clusters, not the Li ions by themselves, diffuse to and react on the Li metal electrode to form the SEI layer. On top of that, more solvent decomposition than salt decomposition favorably occurs under the dilute electrolyte concentration. This is because the lowest unoccupied molecular orbital (LUMO) level of the solvent molecules locates lower than that of the salt anions[52]. Thus, the solvent selection directly dictates the characteristics of the SEI layer on the Li metal. Solvents with high-DNs such as DMSO, DMA, and DMF were previously reported to establish a thin and chemically vulnerable SEI, which can easily be destroyed by the electrophilic attack of solvent molecules, leading to continuous decomposition of the metal[53,54]. For the electrolytes with DMSO, DMA, and DMF solvents, both the impedances and the polarizations in Fig. 5 increased due to interfacial instability with Li metal anodes. On the other side, especially with DOL in DOL:DME mixture, a partially polymerized stable SEI layer formulates on the anode[55]. The chemically durable SEI from DOL:DME enabled the salt-modified electrolytes to maintain good compatibility under different salt conditions. Above 250 cycles, however, the LiBr electrolyte exhibited voltage fluctuations possibly caused by gradual electrode/electrolyte degradation. Thus, further studies are needed to improve the electrolyte stability using high-DN salt anions.

**High sulfur utilization for practical sulfur cathodes**. To demonstrate the generality of the strategy, a freestanding carbon nanotube (CNT) electrode was prepared. Notwithstanding the CP electrode can clearly exhibit the electrochemical effect and deposition morphology of 3D $Li_2S$ growth, its high areal mass ($4.1\,mg\,cm^{-2}$) deteriorates the gravimetric energy density and discourages the application in a practical Li–S cell[56]. Therefore, CNT was employed to build a lightweight freestanding electrode ($1.9\,mg\,cm^{-2}$) and to verify whether the high-DN salt anions maintain their role with the high-surface-area electrode. Due to the increased carbon surface area, the CNT electrode with the conventional LiTFSI electrolyte showed a reasonable discharge capacity (Supplementary Fig. 11); nevertheless, it did not achieve high sulfur utilization due to the electrode passivation by insulating $Li_2S$ film. In comparison with the LiTFSI electrolyte, the LiTf and LiBr electrolytes exhibited extended discharge capacities from the lower discharge plateau. The capacity of the CNT cell with the LiBr electrolyte was 86% of the theoretical capacity ($1449\,mA\,h\,g^{-1}$) at 0.2 C. Due to the smaller interspace volume of the CNT electrode, which may impede 3D $Li_2S$ growth, the specific capacity with the CNT electrode was slightly lower than with the CP electrode. However, the role of the high-DN anions

in delaying electrode passivation was maintained with the CNT electrode as well.

In addition, a high sulfur loading ($3\,mg\,cm^{-2}$) Li–S battery was tested using the LiBr electrolyte to examine its possibility for commercial usages. As shown in Supplementary Fig. 12, an impressive first discharge capacity of $4.53\,mA\,h\,cm^{-2}$ ($1510\,mA\,h\,g^{-1}$) was achieved, which verifies that the high-DN salt anion still maintained its effective role with the high sulfur loading. The obtained areal capacity is practically meaningful because the areal capacity $>4\,mA\,h\,cm^{-2}$ is conventionally regarded as the minimum requirement for designing commercial Li–S batteries[57,58]. In addition, a reversible charge/discharge operation was achieved for succeeding cycles. Recall that the extended capacity obtained from the salt modification strategy has an analogy in the $Li_2O_2$ electrodeposition studies. Superoxide intermediate radicals ($O_2^{\bullet-}$) can be stabilized in the electrolytes using high-DN solvents[59,60] or highly associated salts[61,62]. Subsequently, the solution-mediated pathway results in large toroidal particles of $Li_2O_2$ and augments the initial discharge capacity of the cell. However, the biggest pitfall of the 3D toroidal $Li_2O_2$ is poor reversibility caused by the large charging overpotential. Crystalline $Li_2O_2$ toroid particles are not easily decomposable during charging, which causes gradual electrode passivation with additional cycles[63,64]. Distinct from the Li–air case, in Li–S batteries, various PS anions dissolved in the high-DN electrolytes actively participate in the chemical decomposition of $Li_2S$ during the charging process[5]. UV–Vis absorption spectra of the three catholytes display a difference in solvated PS anion amounts as the supporting salt anions change (Supplementary Fig. 13). The absorbance peaks of $S_8^{2-}$ and $S_3^{\bullet-}$ radicals (at 560 and 617 nm[32], respectively) were increased with the electrolytes of higher-DN anions. Moreover, the overall solvated amount of all PS anions surged higher with the LiBr electrolyte. These PS anions are known to behave like a redox mediator for decomposing $Li_2S$; therefore, the oxidation of large $Li_2S$ 3D particles can be accelerated under the LiTf and LiBr electrolytes. The result clearly supports the smaller charging polarizations with the high-DN anions, as well as the recovery of carbon surfaces and impedances after the charging of the high-DN systems.

**Amended growth pathway with high-DN anions**. Understanding the $Li_2S$ deposition mechanism, changed by the salt property, offers guidance for further advancement on electrolyte design. Possible reasons for the lower plateau extension with controlling electrolytes have been proposed by multiple pioneering reports. Cuisinier and co-workers proved that high-DN electrolytes can stabilize $S_3^{\bullet-}$ radicals, which facilitate chemical redox reactions with other sulfur species including sulfide ($S^{2-}$) anions[33]. However, population dominance of the radical $S_3^{\bullet-}$ is reported to be relatively low during the lower plateau reaction[65]. Thus, the presence of $S_3^{\bullet-}$ radical alone cannot fully explain the enhanced lower plateau capacity of our high-DN salt system. In addition, Pan and co-workers recently suggested that 3D $Li_2S$ nucleation and growth were favorably induced with increasing $Li^+$ diffusion coefficient. $Li^+$ diffusivity affects the morphology of initial nucleation of $Li_2S$, the scale of which does not hamper the electron transfer through the $Li_2S$ nuclei. However, 3D $Li_2S$ growth up to micron-scale (~30 μm) that we examined from the high-DN electrolytes is in a different regime. Since the electrochemical reaction is less likely to occur on the surface of the large-sized $Li_2S$ depositions owing to its extremely low conductivity (~$10^{-13}\,S\,cm^{-1}$)[66], the influence of $Li^+$ diffusion on the particulate $Li_2S$ growth would be less significant.

According to previous research on Li–air batteries, the morphology of $Li_2O_2$ varies depending on the solubility of a

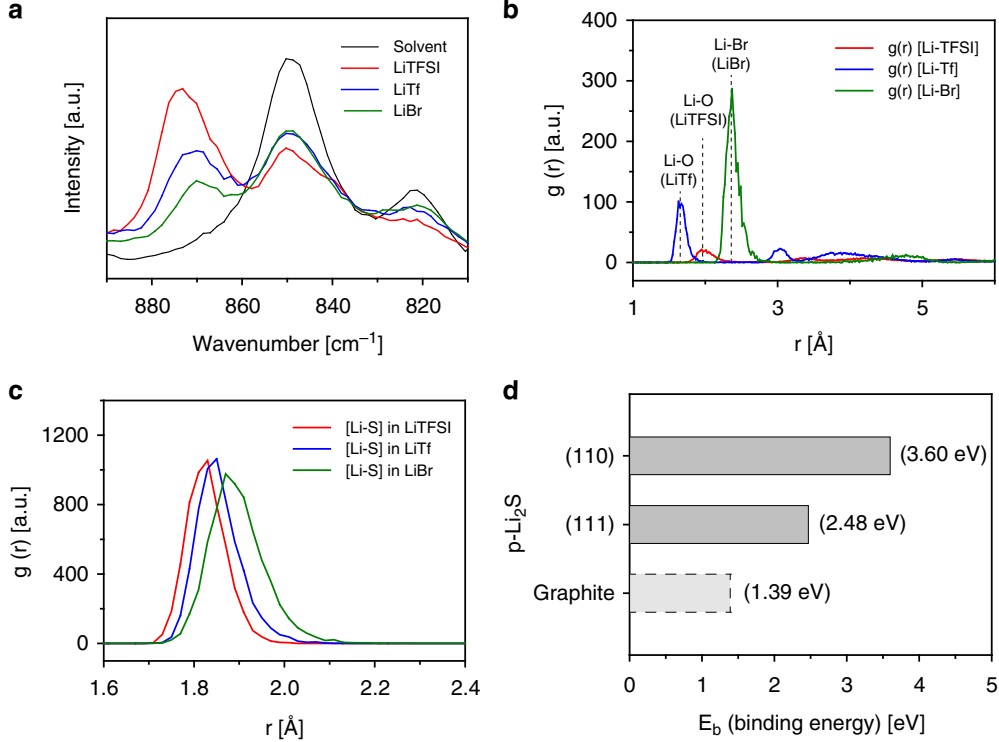

**Fig. 6** The effect of electrolyte anions on lithium sulfide solubility. **a** Raman spectra of 1 M LiX, X = bistriflimide (TFSI⁻), triflate (Tf⁻), or bromide (Br⁻), supporting salts in 1,3-dioxolane (DOL):1,2-dimethoxyethane (DME) (1:1) solutions with respect to the pure solvent mixture (black). **b** Radial distribution functions (how density ($g(r)$) varies as a function of radial distance ($r$)) of X⁻ anions from lithium ion (Li⁺) under the 1 M LiX in DOL:DME solutions. **c** Radial distribution functions of lithium cation–sulfide anion (Li⁺–S²⁻) in different salt environments. **d** Summary of the binding energy of newly generated Li₂S on different surfaces (graphitic carbon surface vs. Li₂S precipitates (p-Li₂S)) based on the first-principle calculations

superoxide intermediate radical (O₂•⁻) [59,61]. Similarly, we expect that the 3D Li₂S growth with the high-DN anions would be originated from enhanced solubility of S²⁻ anions. It is commonly assumed that Li₂S exists more as an ion pair than as a dissociated state under aprotic solvent conditions. However, as Cuisinier and co-workers suggested, a high-DN solvent can cause a surge of partially dissociated Li₂S in the electrolyte [33]. The effect of a salt anion on Li₂S solubility can be estimated using the common ion effect [67]. Regarding a system including a Li salt (LiX, X = TFSI, Tf, or Br) and Li₂S, two competing reactions related to Li⁺ dissociation can be expressed using the equilibrium equations (Eqs. (1) and (2)),

$$\text{LiX} \leftrightarrow \text{Li}^+ + \text{X}^- \quad K_{d1} = \frac{[\text{Li}^+][\text{X}^-]}{[\text{LiX}]}, \tag{1}$$

$$\text{Li}_2\text{S} \leftrightarrow 2\text{Li}^+ + \text{S}^{2-} \quad K_{d2} = \frac{[\text{Li}^+]^2[\text{S}^{2-}]}{[\text{Li}_2\text{S}]} \tag{2}$$

where $K_{d1}$ and $K_{d2}$ are the dissociation equilibrium constants of the LiX and Li₂S, respectively. By combining the two equations, the concentration of solvated S²⁻ anions, [S²⁻], is derived as the form of Eq. (3).

$$[\text{S}^{2-}] = \frac{[\text{Li}_2\text{S}][\text{X}^-]^2}{[\text{LiX}]^2} \cdot \frac{K_{d2}}{K_{d1}^2} \tag{3}$$

The dissociation constant of LiX ($K_{d1}$) is presumed to be smaller when the X⁻ anion has a high electron donating ability. This is because a high-DN anion tends to more strongly associate with a Li⁺ ion according to the hard and soft acids and bases (HSAB) theory. Hence, LiTf and LiBr are expected to have lower ionic dissociation constants ($K_{d1}$) than LiTFSI.

The dissociation of the three LiX salts were monitored by Raman spectroscopy as shown in Fig. 6a. The salt-free DOL:DME mixture showed two band peaks at 850 and 820 cm⁻¹ that are ascribed to the free ethylene oxide groups of solvent molecules [68,69]. The peak intensities were reduced as adding a salt because the solvent molecules coordinate with Li⁺ to form solvation clusters. However, among the three electrolytes, the free solvent peaks diminished less with the LiTf and LiBr electrolytes than with the LiTFSI one. This indicates that the larger number of solvent molecules remained uncoordinated under the high-DN salt anions since the Li⁺ ions bind favorably with the anions instead of with the solvents. Moreover, a newly generated peak at nearby 870 cm⁻¹ is assigned to the signal from the coordinated solvent molecules to Li⁺, thus reflecting the quantity of dissociated Li⁺ from the salt anions [68]. The comparison of the peak intensities at 870 cm⁻¹ shows a tendency that the higher DN the anion has, the less the intensity rises. Therefore, the dissociation amount of Li⁺ decreased in the order of LiTFSI > LiTf > LiBr. Both of the Raman signal changes confirm that the electrolytes with the high-DN anions have lower values of $K_{d1}$. Because the concentration of free S²⁻ anions is inversely proportional to the $K_{d1}$ according to Eq. (3), dissociation of Li₂S (in other words, solubility of Li₂S) would be enhanced with the use of the high-DN anion salts.

In conjunction with the Raman spectroscopy, classical molecular dynamics (MD) simulations were conducted to provide the molecular scale evidence of salt dissociation and its effect on the Li₂S solubility. First, to affirm the difference in the anion

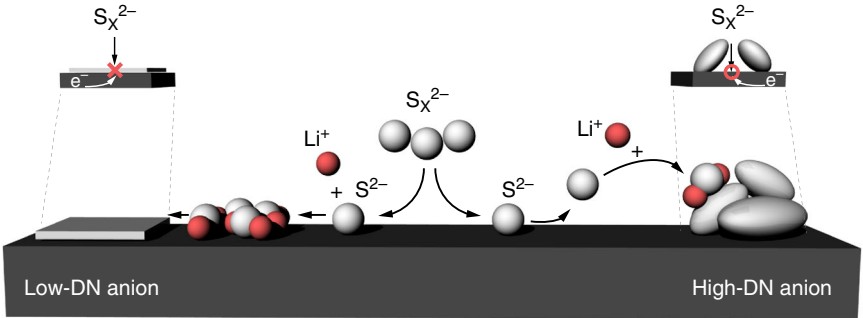

**Fig. 7** Proposed mechanisms for the different lithium sulfide growth behaviors. Based on the solubility of lithium sulfide ($Li_2S$) in an electrolyte medium, high-donor number (DN) anions can induce the three-dimensional growth of $Li_2S$, effectively delaying the electrode passivation

dissociation, the ionic interactions between $Li^+$ and supporting salt anions were examined by calculating the radial distribution functions (RDF) of $Li^+$-$TFSI^-$, $Li^+$-$Tf^-$, and $Li^+$-$Br^-$ in DOL: DME (1:1) (Fig. 6b). As examined from the Li-O interaction of $Li^+$-$TFSI^-$ and $Li^+$-$Tf^-$, $Tf^-$ anions bind at a closer distance from $Li^+$ ions (1.63 Å) than $TFSI^-$ (1.97 Å), indicating the strong attraction between $Li^+$ and $Tf^-$ compared to another. The population of closely existing $Tf^-$ anions also increased based on the intimate relationship between the cation and the high-DN anion. At the same time, although the direct comparison of the binding distance is difficult due to the larger size of a $Br^-$ anion, the strongest $g(r)$ intensity was observed for the $Li^+$-$Br^-$ case. The RDF differences of the three salt-modified electrolytes prove that the anion with higher-DN more strongly associates with $Li^+$. Additionally, fewer DME molecules were observed adjacent to $Li^+$ for the electrolytes using $Tf^-$ and $Br^-$ (Supplementary Fig. 14). This also indirectly supports stronger binding between $Li^+$ and the high-DN anions, because the increased cation–anion interaction would interrupt the attraction between $Li^+$ ions and DME molecules. Therefore, the MD simulation results closely correspond to the ionic dissociation tendencies, observed from the Raman spectra in Fig. 6a.

Then, as the common ion effect provides, we expect that the less dissociating high-DN anions would enhance the partial solubility of $Li_2S$, compensating the total $Li^+$ concentration in the electrolytes. The RDFs between $Li^+$ and $S^{2-}$ were computed for the electrolytes of 0.2 M $Li_2S$ and 1 M LiX (X = TFSI, Tf, or Br). Despite the interactive chemical equilibration of various PS species, their interactions with salt or solvent molecules are independent from each other[70]. Thus for simplicity, only $S^{2-}$ anion species was taken into account. The function of $Li^+$-$S^{2-}$ in the low-DN anion electrolyte depicts a strong affinity between the cation and the $S^{2-}$ anion (Fig. 6c), resulting in extremely low solubility of $Li_2S$ in 1 M LiTFSI DOL:DME (1:1)[70]. In sharp contrast, weakening of the $Li^+$-$S^{2-}$ interaction was observed with increasing the DN of a salt anion. The increase in the $Li^+$-$S^{2-}$ binding distance and the decrease in the closely existing $S^{2-}$ population in Fig. 6c prove that a $S^{2-}$ anion and a supporting salt anion compete one on one for binding with $Li^+$. This may lead to an increase in the partial solubility of $Li_2S$ in the high-DN electrolytes. The effect of a salt anion on the solubility of sulfur species was once reported in the work on LiPS flow batteries. Using LiTf as a supporting salt of the flow electrolyte in combination with DMSO solvent, Pan and co-workers obtained the enhanced solubility of all LiPS species including $Li_2S_2$, a solid-state intermediate[71]. Consistent with the previous finding, our results from the high-DN anion-based electrolytes displayed similarity in enhanced partial solubility of $Li_2S$.

Based on the spectroscopic and computational analyses, we suggest that the high-DN salt anions can increase the partial solubility of $Li_2S$ in the electrolyte. Then, how would these mobile $S^{2-}$ anions induce the 3D growth of $Li_2S$ precipitates during the discharge reaction? In the conventional low-DN electrolyte system, which cannot retain enough solubility of $Li_2S$, $Li_2S$ molecules directly deposit on a carbon site where the reduction occurs. However, when $S^{2-}$ anions can freely move in an electrolyte phase even after the complete reduction, $Li_2S$ would rather deposit on the surface of the other $Li_2S$ precipitates because of the high polarity. To validate the concept, $Li_2S$ binding energy on the two potential deposition sites, a carbon surface and a surface of precipitated $Li_2S$ (p-$Li_2S$), were calculated. Only the chemical binding energy was considered because of the relative dominance of chemisorption of sulfur species compared to the physical adsorption[72]. As suggested in Fig. 6d and Supplementary Fig. 15, the binding energy of $Li_2S$ on the carbon interface (1.39 eV) was much lower than the values on the p-$Li_2S$ surfaces regardless of the facets (3.60 and 2.48 eV, respectively). These strong bindings on p-$Li_2S$ are analogous to those of other metal sulfides such as $CoS_2$, $Co_9S_8$, or FeS, which are known to bind sulfur species based on their high polarity[73,74]. The energy differences suggest that carbon fibers of the CP electrode form weaker attractions with the $Li_2S$ molecules dissolved in the electrolyte, whereas $Li_2S$ precipitates induce strong adsorption of the free $Li_2S$ molecules. Hence, the growth of $Li_2S$ proceeds in a way to build $Li_2S$ agglomerates when the solubility of $Li_2S$ exceeds a certain limit, leading to 3D growth of $Li_2S$.

Using the MD simulation that correlates with Raman spectroscopy and the binding energy calculation, we demonstrate that 3D growth of $Li_2S$ in the high-DN anion electrolytes can be originated from the enhanced solubility of $Li_2S$ and stronger adsorption of the species on the p-$Li_2S$ surfaces. The overall mechanisms for the different $Li_2S$ deposition morphology depending on the DN of salt anions are illustrated in Fig. 7. The soluble $Li_2S$ is presumed to be the key player for the 3D growth of $Li_2S$, which ultimately enables a dramatic increase of active mass utilization during discharge. In case of a salt anion with low-DN, represented by LiTFSI, $Li_2S$ film formation through a surface-mediated pathway is triggered due to the limited $Li_2S$ solubility. Then, the surface is acutely passivated by the film-like $Li_2S$ deposits. On the other hand, a high-DN anion salt such as LiTf or LiBr derives the $S^{2-}$ dissociation in an electrolyte. The $S^{2-}$ anions can travel away from the electrode surface and, after the complete binding with $Li^+$, $Li_2S$ deposits on the top surface of nearby $Li_2S$ agglomerates, resulting in 3D $Li_2S$ growth. The 3D growth of $Li_2S$ can enable the notable extension of the lower voltage plateau and high discharge capacity close to the theoretical value.

## Discussion

By controlling the electron donating property of an electrolyte salt anion, different growth trajectories of $Li_2S$ were generated. The anions with high-DN preferentially induced 3D particle-like growth of $Li_2S$, while the low-DN anion resulted in a film-like morphology. The 3D growth in the high-DN anion systems effectively delayed electrode passivation, and consequently led to high sulfur utilization of 92% even with the carbon host having an extremely low surface area. In spite of the formation of large $Li_2S$ particles for electrolytes with the high-DN anions, the $Li_2S$ agglomerates were readily decomposed while charging due to the redox mediation of LiPS with $Li_2S$, achieving high coulombic efficiency. The different $Li_2S$ morphologies, controlled by the salt anions, were explained in terms of the difference in $Li_2S$ solubility. In contrast to the electrolytes with high-DN solvents, the electrolytes with high-DN salt anions showed incomparably better compatibility with a Li metal electrode, which allows a stable cycling of the corresponding Li–S batteries. Nevertheless, the approach needs to be complemented to provide better Li metal stability for an extended cycle life. We believe that this contribution yields a simple but novel strategy for designing high-capacity Li–S batteries through controlling the intrinsic deposition chemistry of $Li_2S$.

## Methods

**Catholyte preparation.** LiPS ($Li_2S_8$ based) solutions of 0.2 M were prepared by heating and stirring stoichiometric amounts of lithium sulfide ($Li_2S$) and sulfur ($S_8$) (both from Sigma-Aldrich) in DOL:DME (Sigma-Aldrich) (1:1 in volume). The catholytes were mixed at 60 °C for 12 h, along with 0.2 M lithium nitrate ($LiNO_3$, Sigma-Aldrich) additive and one of the following lithium salts: 1 M lithium bis (trifluromethanesulfonyl)imide (LiTFSI, 3 M), lithium trifluoromethanesulfonate (LiTf, Sigma-Aldrich), or LiBr (Sigma-Aldrich).

**Electrochemical measurements.** Electrochemical performance of each electrolyte was evaluated using a 2032-type coin cell. A 14 pi of CP (TGP-H030, Toray) with a surface area of $0.9904 \, m^2 \, g^{-1}$ was inserted as a working electrode. A 16.5 pi of lithium foil (450 μm thickness, Honjo Metal) was used as a counter electrode, and an 18 pi of Celgard 2400 membrane was used as a separator. A portion (30 μL, corresponding to 1 mg cm$^{-2}$ areal sulfur loading) of three different salt (LiTFSI, LiTf, and LiBr) based LiPS electrolytes were used for each.

A freestanding CNT (LG Chem.) electrode was fabricated by mixing 25 mg of CNT with 15 mg of polyvinylpyrrolidone (PVP-40, Sigma Aldrich) as a surfactant in 50 mL of ethanol (Merck). The mixed solution was tip-sonicated for 30 min and then vacuum-filtered to build a freestanding electrode. The obtained electrode was dried at 60 °C for 12 h to remove the residual solvent. A 2032-type coin cell was assembled with the same components as used for the electrochemical cell test: a 16.5 pi Li metal anode, an 18 pi Celgard separator, and 0.2 M LiPS catholyte, except for replacing the positive electrode with the CNT electrode. A 3 mg cm$^{-2}$ sulfur-loaded cell was assembled using 90 μL of the LiBr-based catholyte, with maintaining the same carbon/sulfur areal ratio. All cells were assembled in an Ar-filled glove box and were operated using a TOSCAT-3000U (Toyo System) within a voltage range of 1.8–2.7 V.

Three-electrode (3-electrode) EIS was performed using a Solartron 1470E Frequency Response Analyzer (Solartron Analytical) in a frequency range from 1 MHz to 0.1 Hz, with a perturbation degree of 10 mV. The cathode impedances were recorded separately using a 3-electrode pouch-type cell configuration, which employed a Li metal reference electrode (Supplementary Fig. 5).

The stability of Li metals under different electrolyte compositions was verified using a Li–Li symmetric cell condition. A 16 pi and a 12 pi of lithium foil (150 μm thickness, Honjo Metal) were used as a positive and a negative electrodes, respectively, and an 18 pi of a Glass Microfiber Filter (GF3 grade, CHMLAB GROUP) was used as a separator. For the Li metal stability test cell, 100 μL of each of the six solvent or salt-modified electrolytes was used.

**Chronoamperometry analysis.** The freestanding CNT electrode was used instead of the CP, to provide enough surface area to examine the electrochemical growth behavior during discharge. Test cells were assembled using a 16.5 pi Li metal anode, an 18 pi Celgard separator, a free-standing CNT electrode, and 0.2 M LiPS catholyte. CV was conducted in advance of CA with a scan rate of 0.1 mV s$^{-1}$ to determine the test voltage for the CA analysis. Cells with electrolytes of different salt anions were initially discharged at 2.2 V for 6 h, or until the current fell below 0.01 mA. Then, a voltage of 2.0 V was applied for an additional 3 h to induce a $Li_2S$ electrodeposition reaction. The current behaviors of the cells during the

potentiostatic discharge were recorded using a VSP Potentiostat System (Bio-logic) during the CA analysis.

**Electrode and electrolyte characterization.** SEM analysis of pristine CP and discharged cathodes was performed using a Sirion Field-Emission Scanning Electron Microscope (FE-SEM, Sirion, FEI). Surface characterization of discharged cathodes was conducted using an X-ray photoelectron spectroscope (XPS; K-alpha, Thermo VG Scientific) with Al Kα as the X-ray source. The binding energies obtained from XPS analysis were calibrated based on the hydrocarbon C 1s peak at 284.7 eV. XRD of the discharged electrode samples was conducted on a High-Resolution Powder X-Ray Diffractometer (Smartlab, RIGAKU) with a Cu Kα radiation source at a scan rate of 5° min$^{-1}$. The discharged electrodes for SEM, XPS, and XRD measurements were neatly rinsed with DME, and then were dried under vacuum condition to eliminate residual soluble salts and solvents.

Raman spectra of electrolytes with different supporting salts were collected with a Dispersive Raman Spectrometer (ARAMIS, JY Horiba) using a 514 nm wavelength laser. A small portion (1 mL) of each electrolyte solution was taken into a glass capillary tube for the liquid Raman spectra measurement. The obtained spectra were normalized using the highest intensity peak between 1400 and 1550 cm$^{-1}$, which is assigned to the $CH_2$ bending/scissoring mode.

UV–Vis absorption spectroscopy was performed using a UV–Vis Spectrophotometer (GENESYS 10S, Thermo Scientific). Diluted LiPS catholytes (1 mM), with the three supporting salts, were placed in 10 mm High Precision Cells (Hellma Analytics) and were measured within the wavelength range of 300–800 nm.

**Computational simulations.** MD calculations were performed using the Material Studio (BIOVIA, 2018) software package. Geometries of DOL, DME, $Li_2S$, LiTFSI, LiTf, and LiBr molecules were optimized using the Forcite module and Forcefield COMPASS II, as provided in the Material Studio databases. Ionic dissociations of LiTFSI, LiTf, and LiBr in DOL:DME electrolytes were estimated by placing 357 DOL, 241 DME, 50 Li$^+$, and 50 X$^-$ (X$^-$ = TFSI$^-$, Tf$^-$, or Br$^-$), the ratio of which corresponds to that in the 1 M Li salt in DOL:DME (1:1). The systems were geometrically stabilized using Smart Algorithm employing a convergence tolerance of 0.001 kcal mol$^{-1}$ Å$^{-1}$. Then, the systems were equilibrated in NPT and NVT ensembles using the NHL Algorithm with a Q ratio of 0.01 and a decay constant of 0.1 ps to control the pressure and temperature of the cell. After the systems were stabilized with a geometrical optimization process, 3 ns of MD simulations were conducted at 450, 363, and 298 K NPT at 1 atm. Subsequently, the systems were run for 1 ns at 298 K in an NVT ensemble[75], and the radial distribution functions (RDFs) were collected based on the last stabilized system. The solubility of S$^{2-}$ anions in different salt anion environments was examined simply by placing additional 20 Li$^+$ and 10 S$^{2-}$ molecules to the as-prepared electrolyte simulation cell.

The binding energies of a $Li_2S$ molecule on a carbon surface and pre-deposited $Li_2S$ were calculated with the Density Functional Theory (DFT) calculation method using the same analytical software. Electron exchange correlations were described with the Perdew–Burke–Ernzerhof Generalized Gradient Approximation (GGA-PBE), and for considering the Van der Waals interaction, a semi-empirical dispersion potential in the DFT-D method of Grimme was used. Double numerical plus polarization (DNP+) was used as the basis set. Among the calculations, the maximum value of the energy, force, and displacement were set to $1.0 \times 10^{-5}$ Ha (hartree), $2 \times 10^{-3}$ Ha Å$^{-1}$, and $5 \times 10^{-3}$ Å, respectively[76]. The corresponding k point grid is generated by the Monkhrost–Pack technique for the Brillouin zone sampling and the sampling was carried out using a $5 \times 5 \times 1$ grid[76,77]. For the binding energy calculation, supercell structures of the selected facets of $Li_2S$ and carbon layer were constructed and 15 Å vacuum is applied to eliminate the influence of another slab arising from the periodic boundary conditions. The binding energy values of $Li_2S$ on the surfaces were calculated by obtaining the total energy differences between a $Li_2S$ adsorbed surface and a pristine surface with an unbound $Li_2S$ molecule. Snapshots of the binding energy calculation cells are provided in Supplementary Fig. 15.

## Data availability

The data that support the findings in this study are in the published article and/or its Supplementary Information files. The whole datasets are available from the corresponding author on reasonable request. The source data underlying Fig. 4b are provided as a Source Data file.

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

## Acknowledgements

This work was supported by LG Chem., the KAIST Institute for Nano-Century (KINC), and by Technology Development Program to Solve Climate Changes through the National Research Foundation of Korea (NRF) funded by the ministry of Science, ICT (2018M1A2A2063807).

## Author contributions

H.C., H.N., and H.-T.K. conceived the concept of 3D $Li_2S$ growth with high-DN salt anions and designed this work. H.C. carried out the experimental planning, electro-chemical measurements, and data analysis; H.N. performed the characterization of materials and computational simulations including MD and DFT calculations; H.C. and Y.J.K. proposed the reaction mechanisms; S.Y. and J.H.L. assisted in the CA analysis; J.L., H.K., Y.K.K., and D.K.Y. contributed to experimental design and discussion of the results; H.C. and H.-T.K. wrote the manuscript, and H.-T.K. supervised this work; all authors commented on the manuscript.

## Additional information

**Competing interests:** The authors declare no competing interests.

