## [Peer Review File · Nature Communications]

Reviewers' Comments:

Reviewer #1:

Remarks to the Author:

This paper reports a new strategy to address Li-S battery technical challenges. The research was well designed and the paper was written/organized very well. It can be published. Some questions may be helpful for the authors to improve their work:

1. The anion seems to affect the solubility of Li₂S. does it affect other solution properties such as the mobility/diffusivity of Li⁺ which may affect Li₂S growth.
2. The electrolyte amount used in the cells seems high, how does the Li-S cell perform if limited amount of electrolyte is used? If use the electrolyte to sulfur ratio, i.e., the ratio <5, preferably <3.
3. In Fig. 1a, cell performance with LiTFSI seems extremely poor. did you get a decent baseline?

Reviewer #2:

Remarks to the Author:

The investigation and research of new electrolyte systems are of particular interests and importance for high-performance secondary batteries such as lithium-sulfur (Li-S) batteries. The manuscript entitled NCOMMS-18-22694 presents very comprehensive investigation of electrolytes with high-Donor Number (DN) anions for lithium-sulfur batteries and clearly demonstrates the effect of high DN anions on enabling three-dimensional (3D) growth of lithium sulfide (Li₂S). A large number of electrochemical measurements (galvanostatic charge/discharge, potentiostatic chronoamperometry, electrochemical impedance, etc.) and analytical tools (ex-situ electron microscopy and X-ray photoelectron spectroscopy (XPS), etc.), as well as theoretical simulations (molecular simulation and binding energy calculation), was employed to unravel the fact that high-DN anions, especially the bromide ion (Br⁻), improves the sulfur utilization to > 90% through preventing the early passivation. This study provides an alternative approach for improving the energy density of practical Li-S batteries and opens up new avenues toward rational design of high-performance Li-S batteries. The mechanistic insights into the dissolution and deposition behaviors of Li₂S are also interesting and helpful. In general, I would like to suggest its publication on Nature Communications. But there would be space for improvement with respect to the conceptual generality, mechanistic unequivocality, and result reproducibility.

1. Most of the contents are about the promotion of 3D Li₂S growth. The stability of lithium metal anodes with LiBr/DOL:DME is not as good as LiTFSI/DOL:DME (Fig. 5g) although high-DN anions are much more stable against lithium than high-DN solvents. The authors also agree with that "further studies are needed to improve the stability of high-DN electrolytes." (Line 304-305). Therefore, to prevent misleading, I would not suggest to highlight the lithium metal stability in the title as high-DN anions are basically compared to the routine TFSI⁻ anion.
2. The authors employed two different current collectors: low-surface-area carbon paper (CP) and high-surface-area carbon nanotube (CNT) paper. I am wondering if the effect of anion DN is less dominant in the case of CNT paper than that of CP. The reasons include: (1) the cyclic voltammograms (CVs) with CNT paper (Fig. 7) suggest different discharge behaviors from that indicated in Fig. 1 with CP; (2) CP is much heavier in areal density (~4-5 mg cm⁻² for TGP-H-030) than CNT paper, which is detrimental to the sulfur weight fraction (as demonstrated in a previous literature of J. Am. Chem. Soc. 2017, 139, 8458, the sulfur fraction can be as high as 60%). Cycling performance similar to that presented in Fig. 1 is thereby suggested to be provided with CNT current collectors.
3. The origin of why high-DN anions prevent the early passivation is slightly equivocal. From the XPS results (Fig. 3c), the lack of Li₂S peaks for the sample discharged with the LiBr electrolyte implies the absence of deposited Li₂S. Therefore, the high solubility of Li₂S is likely the reason as soluble Li₂S clusters in the LiBr electrolyte can promote the 3D growth kinetically (Nature Energy 2017, 2, 813).

But from the UV-Vis results (Supplementary Fig. 10), the author also observed the presence of $S_3^{\bullet-}$ radicals, which may probably alter the discharge reaction pathways by delaying the production of Li_2S intrinsically (Adv. Energy Mater. 2015, 5, 1401801). For the above kinetic and intrinsic reasons, which one would be the most likely fundamental reason? Please point it out clearly with reasonable data support.

4. From Line 419–437, the authors attribute the origin of promoted 3D growth of Li_2S to stronger binding energy of Li_2S to the surface of precipitated Li_2S than that of carbon. However, the difference in binding energies cannot explain the preference of 3D growth within electrolytes with high-DN anions because Li_2S nuclei also form in LiTFSI/DOL:DME electrolyte to adsorb Li_2S clusters. Such an adsorption on pre-existed Li_2S should be a common phenomenon for all electrolytes. In my opinion, the diffusivity of Li_2S would account for the observed difference in deposition behaviors. It is suggested to measure the Li^+ diffusion coefficients in different electrolytes as done in a previous literature (Nature Energy 2017, 2, 813).

5. Although the electrochemical measurements are quite comprehensive and solid, tests with more cycle numbers and under higher current densities are required to examine the long-term serviceability and rate capability. Besides, the potential of these electrolytes with high-DN anions for practical Li–S batteries operated at a lean-electrolyte condition is suggested to be commented on.

Response Letter

Title: Electrolytes with high Donor Number anions: Achieving high sulfur utilization of Li-S batteries with three-dimensional Li_2S growth

We highly appreciate for peer reviewing our manuscript and giving helpful and critical comments and advices. The constructive comments from the reviewers have significantly improved the quality of this manuscript. We revised the manuscript reflecting reviewers' advices and corrected some errors. We hope that this revision can fulfill the reviewers' comments. The changes made are indicated by yellow background in the revised manuscript.

Reviewer #1 (Remarks to the Author):

1. The anion seems to affect the solubility of Li_2S . Does it affect other solution properties such as the mobility/diffusivity of Li^+ , which may affect Li_2S growth?

Fig. R1 | Cyclic Voltammograms (CV) of the electrolytes with different salt anions: (a) LiTFSI, (b) LiTf, and (c) LiBr with varying scan rates. Plots of the CV peak currents of (d) the second cathodic reaction (C2, $\text{Li}_2\text{S}_4 \rightarrow \text{Li}_2\text{S}$), (e) the first anodic reaction (A1, $\text{Li}_2\text{S} \rightarrow \text{Li}_2\text{S}_4$) versus the square root of scan rates.

Fig. R2 | Snapshots of lithium ion (Li^+) solvation cluster with different salt anions: **(a)** the LiTFSI electrolyte and **(b)** the LiBr electrolyte.

Response: We highly appreciated the reviewer's comments for our manuscript. It was previously reported that lithium ion (Li^+) diffusion coefficient can influence the nucleation morphology of Li_2S (Nature Energy 2017, 2, 813). As suggested, Li^+ diffusivity was investigated for the three electrolytes. Fig. R1 displays the Cyclic Voltammetry (CV) curves of the electrolytes collected at different scan rates (0.05 to 0.25 mV s^{-1}). There exist two cathodic peaks at around 2.30 V (C1) and 2.00 V (C2), corresponding to the reduction of elemental sulfur (S_8) to long-chain lithium polysulfides (LiPSs) and the successive reduction to short-chain Li_2S , respectively. Also, the two anodic peaks at around 2.30 V (A1) and 2.50 V (A2) are from the decomposition of solid Li_2S to LiPSs and further oxidation to S_8 , respectively. The second cathodic (C2) and the first anodic (A1) current peaks, relevant to the reduction and oxidation of Li_2S , show a linear relationship with the square root values of scan rates. Then, the slopes of the curves represent the Li^+ diffusion rate where other environments are equal (PNAS 2017, 114(5), 840).

As provided in Fig. R1, the LiTFSI electrolyte showed the smallest slope of the I_p vs. $v^{1/2}$ curve, followed by the LiBr and LiTf electrolyte in order. Since larger numbers of solvent molecules coordinate with Li^+ under the low-DN anion condition (Fig. 6), the LiTFSI electrolyte exhibits the lowest Li^+ diffusivity. However, Li^+ diffusivity is higher for the LiTf electrolyte than the LiBr electrolyte, which does not directly match with the 3D Li_2S growth trend in our manuscript. The lower diffusivity for the highest-DN anion electrolyte may be due to the extremely high ionic strength of Br^- anion, which triggers an agglomerate form of the Li^+ solvation cluster as expected from the molecular dynamics simulation (Fig. R2). Moreover, unlike the previous work, which dealt with the Li^+ diffusivity effect on submicron-size Li_2S nucleation, the scale of our Li_2S deposition in Fig. 2 exceeds tens of micrometers. Therefore, we suggest that the Li_2S solubility increase would play the predominant role on the 3D growth of Li_2S .

2. The electrolyte amount used in the cells seems high, how does the Li-S cell perform if limited amount of electrolyte is used? If use the electrolyte to sulfur ratio, i.e., the ratio <5 , preferably <3 .

Fig. R3 | Lithium polysulfide (LiPS, Li_2S_8 based) solubility changes of 1 M LiTFSI DOL:DME(1:1) electrolyte with decreasing the electrolyte to sulfur ratio (E/S ratio) of 20:1, 5:1, and 3:1.

Response: We appreciate the comments and agree with the reviewer's proposal. Unfortunately, the catholyte based Li-S system, which we employed to examine the electrochemical effect and the deposition morphology of 3D Li_2S growth, is not suitable to fabricate the low electrolyte/sulfur (E/S) ratio Li-S batteries. It is mainly due to the solubility limit of LiPS species in ether-based electrolytes (J. Electrochem. Soc. 2017, 164(4), A917). Compared to the catholyte sample with the E/S ratio of 20:1, the catholytes with the E/S ratio of 5:1 and 3:1 do not completely dissolve the lithium polysulfides (LiPSs) as shown in Fig. R3, thus cannot be used for Li-S cells. We have acknowledged the inadequacy of the catholyte based system; therefore, we are currently preparing for the follow-up study to investigate the effectiveness of high-DN anions under low E/S ratio conditions, using a different type of Li-S batteries.

3. In Fig. 1a, cell performance with LiTFSI seems extremely poor. Did you get a decent baseline?

Fig. R4 | (a) BET surface areas of CP, CNT, KB, (b) Charge and discharge curves at the first cycle (0.2 C) and (c) the cycling stabilities at 0.2 C (closed circle : charge capacity, open circle : discharge capacity) for the CNT-based Li-S cells with the LiTFSI, LiTf, and LiBr electrolytes. The electrolytes consist of 0.2 M LiPS (Li_2S_8 based) with 1 M Li salts (LiX, X= TFSI, Tf, or Br) / 0.2 M LiNO_3 / DOL:DME (1:1)

Response: We are grateful for what the reviewer comments. We admit that the low discharge capacity of the reference cell is unusual when comparing the values with the other LiTFSI electrolyte based Li-S systems in previous works. The abrupt failure of the LiTFSI sample is resulted from the extremely low surface area of the electrode (Carbon Paper (CP), BET surface: $\sim 1 \text{ m}^2 \text{g}^{-1}$). The surface area of the CP electrode is 230 times smaller than that of carbon nanotube (CNT) and 1300 times smaller than that of Ketjenblack (KB), commonly used as a carbon host material for Li-S batteries. Consequently, the early passivation of the electrode surfaces was instigated when 2D Li_2S deposition initiated. It is intentionally designed to clearly examine the effect of 3D Li_2S growth on the interfacial electrochemical reactions.

However, as reviewer indicated, to eliminate the confusion, a freestanding CNT electrode was used to supplant the CP electrode and tested with the three salt-modulating electrolytes. As provided in Fig. R4, the capacity differentials among the electrolytes were reduced because of the higher active surface area

and smaller interspace volume of the CNT electrode. Nevertheless, the high-DN anions seem to maintain their role to delay the passivation of the interface by inducing 3D Li_2S growth. These data were added to the revised manuscript

Revised manuscript:

(Line 321-337)

To demonstrate the generality of the strategy, a freestanding Carbon Nanotube (CNT) electrode was prepared. Notwithstanding the CP electrode can clearly exhibit the electrochemical effect and deposition morphology of 3D Li_2S growth, its high areal mass (4.1 mg cm^{-2}) deteriorates the gravimetric energy density and discourages the application in a practical Li-S cell⁵⁶. Therefore, CNT was employed to build a lightweight freestanding electrode (1.9 mg cm^{-2}) and to verify whether the high-DN salt anions maintain their role with the high-surface-area electrode. Due to the increased carbon surface area, the CNT electrode with the conventional LiTFSI electrolyte showed a reasonable discharge capacity (Supplementary Fig. 11); nevertheless, it did not achieve high sulfur utilization due to the electrode passivation by insulating Li_2S film. In comparison with the LiTFSI electrolyte, LiTf and LiBr electrolytes exhibited extended discharge capacities from the lower discharge plateau. The capacity of the CNT cell with the LiBr electrolyte was 86% of the theoretical capacity (1449 mAh g^{-1}) at 0.2 C. Due to the smaller interspace volume of the CNT electrode, which may impede 3D Li_2S growth, the specific capacity with the LiBr electrolyte was slightly smaller than the cell with the CP electrode. However, the role of the high-DN anions in delaying electrode passivation was maintained with the CNT electrode.

(Supplementary Fig. 11).

Supplementary Fig. 11 | Electrochemical profiles of the three salt anions with the freestanding CNT electrode (a) A SEM image and (b) the BET isotherm curve of the CNT freestanding electrode. The BET surface area of the CNT electrode was measured to be $230.4 \text{ m}^2 \text{ g}^{-1}$. (c) Charge and discharge curves at the first cycle (0.2 C) and (d) the cycling stabilities at 0.2 C (closed circle: charge capacity, open circle : discharge capacity) for the CNT-based Li-S cells with the LiTFSI, LiTf, and LiBr electrolyte. The electrolytes consist of 0.2 M LiPS (Li_2S_8 based) with 1 M Li salts (LiX , X= TFSI, Tf, or Br) / 0.2 M LiNO_3 / DOL:DME (1:1)

Reviewer #2 (Remarks to the Author):

1. Most of the contents are about the promotion of 3D Li₂S growth. The stability of lithium metal anodes with LiBr/DOL:DME is not as good as LiTFSI/DOL:DME (Fig. 5g) although high-DN anions are much more stable against lithium than high-DN solvents. The authors also agree with “further studies are needed to improve the stability of high-DN electrolytes.” (Line 304–305). Therefore, to prevent misleading, I would not suggest highlighting the lithium metal stability in the title as high-DN anions are basically compared to the routine TFSI anion.

Response: We thank the reviewer for the comments. As the reviewer remarked, our original intention to highlight the Li metal stability on the title was to emphasize the relative stability of the high-DN anion electrolytes compared to high-DN solvent-based Li-S systems. However, we completely agree that the phrase, “lithium metal stability” can mislead the potential readers; therefore, we discarded the expression and revised the title.

2. The authors employed two different current collectors: low-surface-area carbon paper (CP) and high-surface-area carbon nanotube (CNT) paper. I am wondering if the effect of anion DN is less dominant in the case of CNT paper than that of CP. The reasons include: (1) the Cyclic Voltammograms (CVs) with CNT paper (Fig. 7) suggest different discharge behaviors from that indicated in Fig. 1 with CP; (2) CP is much heavier in areal density (~4–5 mg cm⁻² for TGP-H-030) than CNT paper, which is detrimental to the sulfur weight fraction (as demonstrated in a previous literature of J. Am. Chem. Soc. 2017, 139, 8458, the sulfur fraction can be as high as 60%). Cycling performance similar to that presented in Fig. 1 is thereby suggested to be provided with CNT current collectors.

Fig. R5 | Cyclic voltammetry (CV) (0.1 mV s⁻¹) for 1 M (a) LiTFSI, (b) LiTf, and (c) LiBr based 0.2 M LiPS catholytes with freestanding CNT cathodes and lithium metal anodes. From the area of the second cathodic peaks from the cyclic voltammograms, the discharge capacities were calculated for the electrolytes and compared in (d)

Fig. R6 | (a) A SEM image and (b) The BET isotherm curve of the CNT freestanding electrode. The BET surface area of the CNT electrode was measured as a value of $230.4 \text{ m}^2 \text{ g}^{-1}$. (c) Charge and discharge curves of the first 0.2 C cycle, (d) The charge (closed circle) and discharge (open circle) capacities for 50 charge/discharge cycles at 0.2 C. The electrolytes consist of 0.2 M LiPS (Li_2S_8 based) with 1 M Li salts (LiX, X= TFSI, Tf, or Br) / 0.2 M LiNO_3 / DOL:DME (1:1)

Response: We highly appreciate the comments. The reviewer correctly indicates the limitation of the CP electrode for practical application. We firmly agree with the reviewer's point, even though the electrode enabled us to clarify the electrochemical effect and the deposition morphology of 3D Li_2S growth. As suggested, we prepared the CNT freestanding electrode, which 2.2 times lighter than the CP, and conducted electrochemical cell tests to verify whether the high-DN anions are effective in delaying the electrode passivation for the CNT electrode with higher surface area.

For the CV diagrams of the CNT cells (Fig. R5), we measured the areal capacity from the second cathodic peak, which originates from the lower discharge plateau reaction. Even with the CNT electrode, the LiBr electrolyte showed higher areal capacity than the LiTf and LiTFSI electrolytes, maintaining its role of delaying the electrode passivation. In addition, under galvanostatic cycling at 0.2 C for the CNT electrode, the LiBr electrolyte achieved a discharge capacity of 1449 mAh g^{-1} at the first cycle due to the notably extended lower plateau compared with those of the LiTf and LiTFSI electrolytes. Comparing the LiTf and LiTFSI electrolytes, a slightly larger discharge capacity was observed for the LiTf electrolyte. The capacity amelioration trends are in accordance with those observed using the CP electrode. As the reviewer expected, the effect of anions' DN was less dominant with the CNT electrode because of the higher surface area and smaller interspace volume of the electrode; nevertheless, we examined that the passivation-delaying role of high-DN anions was maintained with the higher-surface-area electrode. These new data would be also meaningful to potential readers of our manuscript, thus were added to the revised manuscript.

Revised manuscript:

(Line 321-337)

To demonstrate the generality of the strategy, a freestanding Carbon Nanotube (CNT) electrode was prepared. Notwithstanding the CP electrode can clearly exhibit the electrochemical effect and deposition morphology of 3D Li_2S growth, its high areal mass (4.1 mg cm^{-2}) deteriorates the gravimetric energy density and discourages the application in a practical Li-S cell⁵⁶. Therefore, CNT was employed to build a lightweight freestanding electrode (1.9 mg cm^{-2}) and to verify whether the high-DN salt anions maintain their role with the high-surface-area electrode. Due to the increased carbon surface area, the CNT electrode with the conventional LiTFSI electrolyte showed a reasonable discharge capacity (Supplementary Fig. 11); nevertheless, it did not achieve high sulfur utilization due to the electrode passivation by insulating Li_2S film. In comparison with the LiTFSI electrolyte, LiTf and LiBr electrolytes exhibited extended discharge capacities from the lower discharge plateau. The capacity of the CNT cell with the LiBr electrolyte was 86% of the theoretical capacity (1449 mAh g^{-1}) at 0.2 C. Due to the smaller interspace volume of the CNT electrode, which may impede 3D Li_2S growth, the specific capacity with the LiBr electrolyte was slightly smaller than the cell with the CP electrode. However, the role of the high-DN anions in delaying electrode passivation was maintained with the CNT electrode.

(Supplementary Fig. 11).

Supplementary Fig. 11 | Electrochemical profiles of the three salt anions with the freestanding CNT electrode (a) A SEM image and (b) the BET isotherm curve of the CNT freestanding electrode. The BET surface area of the CNT electrode was measured to be $230.4 \text{ m}^2 \text{ g}^{-1}$. (c) Charge and discharge curves at the first cycle (0.2 C) and (d) the cycling stabilities at 0.2 C (closed circle: charge capacity, open circle : discharge capacity) for the CNT-based Li-S cells with the LiTFSI, LiTf, and LiBr electrolyte. The electrolytes consist of 0.2 M LiPS (Li_2S_8 based) with 1 M Li salts (LiX , X= TFSI, Tf, or Br) / 0.2 M LiNO_3 / DOL:DME (1:1)

3. The origin of why high-DN anions prevent the early passivation is slightly equivocal. From the XPS results (Fig. 3c), the lack of Li_2S peaks for the sample discharged with the LiBr electrolyte implies the absence of deposited Li_2S . Therefore, the high solubility of Li_2S is likely the reason as soluble Li_2S clusters in the LiBr

electrolyte can promote the 3D growth kinetically (Nature Energy 2017, 2, 813). But from the UV-Vis results (Supplementary Fig. 10), the author also observed the presence of $S_3^{\cdot-}$ radicals, which may probably alter the discharge reaction pathways by delaying the production of Li_2S intrinsically (Adv. Energy Mater. 2015, 5, 1401801). For the above kinetic and intrinsic reasons, which one would be the most likely fundamental reason? Please point it out clearly with reasonable data support.

Response: We are not certain whether we can assert which property was the “fundamental” origin of the enhanced lower plateau capacity. As the reviewer indicated, some of the previous works suggested that the presence of $S_3^{\cdot-}$ can alter the discharge pathways and delay the formation of Li_2S precipitation on the electrode/electrolyte interface (Adv. Energy Mater. 2015, 5, 1401801). As far as we understand, due to the complexity of the sulfur redox reaction, it is challenging to clearly distinguish the role of each polysulfide/sulfide anion during the operation, thus is likely that $S_3^{\cdot-}$ may also have a positive effect on realizing the high sulfur utilization of our high-DN systems.

However, when thinking of the most influential species to what we observed: the extension of the lower discharge plateau, we still believe that the high solubility of Li_2S was the “major” reason to enable the amelioration of the cell performance. First, the chemical redox equilibrium of $S_3^{\cdot-}$ radical with Li_2S , suggested in the previous study (Adv. Energy Mater. 2015, 5, 1401801), can be activated when sulfide (S^{2-}) anions are partially dissociated and solvated by the solvent molecules. That may be the reason why the authors emphasized the partial solvation of Li_2S under high-DN solvent electrolytes. On top of that, according to the same article, the concentration of $S_3^{\cdot-}$ radical is shown to decrease during the voltage dropping region (before the lower plateau) and the Li_2S amount accordingly increases. Therefore, at the lower discharge plateau phase, the population of Li_2S would be the most dominant in the electrolytes or the electrode interfaces. These trends in the polysulfide/sulfide anion population during the discharge reaction is also provided by another previous work, suggesting that $S_3^{\cdot-}$ related S_3^{2-} is produced in the upper plateau and the voltage dropping region, then subsequently reduced to S_2^{2-} and S^{2-} in the lower plateau (Phys. Chem. Chem. Phys. 2014, 16, 9344). In conclusion, because of the population dominance of Li_2S during the lower plateau reaction, our observation of the capacity enhancement is thought to be mainly originated from the enhanced solubility of Li_2S . Understanding the underlying chemistry of the sulfur redox reaction is essential but extremely taxing in some sense. We deeply agree that the question from the reviewer’s comment is especially significant and hope that the further studies would be facilitated, thus our suggestions were added to the revised manuscript.

Revised manuscript:

(Line 366-373)

Understanding the Li_2S deposition mechanism, changed by the salt property, offers guidance for further advancement on electrolyte design. Possible reasons for the lower plateau extension with controlling electrolytes have been proposed by multiple pioneering reports. Cuisinier and co-workers proved that high-DN electrolytes can stabilize $S_3^{\cdot-}$ radicals, which facilitate chemical redox reactions with other sulfur species including S^{2-} anions³³. However, population dominance of the radical $S_3^{\cdot-}$ is reported to be relatively low during the lower plateau reaction⁶⁵, thus the presence of $S_3^{\cdot-}$ radical alone cannot fully explain the enhanced lower plateau capacity of our high-DN salt system.

4. From Line 419–437, the authors attribute the origin of promoted 3D growth of Li_2S to stronger binding energy of Li_2S to the surface of precipitated Li_2S than that of carbon. However, the difference in binding energies cannot explain the preference of 3D growth within electrolytes with high-DN anions because Li_2S nuclei also form in LiTFSI/DOL:DME electrolyte to adsorb Li_2S clusters. Such an adsorption on pre-existed Li_2S should be a common phenomenon for all electrolytes. In my opinion, the diffusivity of Li_2S would account for the observed difference in deposition behaviors. It is suggested to measure the Li^+ diffusion coefficients in different electrolytes as done in a previous literature (Nature Energy 2017, 2, 813).

Fig. R7 | Cyclic Voltammograms of the electrolytes with different salt anions: (a) LiTFSI, (b) LiTf, and (c) LiBr with increasing scan rates. Plots of Cyclic Voltammetry (CV) peak currents of (d) the second cathodic reaction (C2, $\text{Li}_2\text{S}_4 \rightarrow \text{Li}_2\text{S}$), (e) the first anodic reaction (A1, $\text{Li}_2\text{S} \rightarrow \text{Li}_2\text{S}_4$) versus the square root of scan rates.

Fig. R8 | Snapshots of lithium ion (Li^+) solvation cluster under different salt anions: (a) the LiTFSI electrolyte and (b) the LiBr electrolyte.

Fig. R9 | (a) Background fitting of current vs. time curve for chronoamperometry (CA) test at 2.00 V (LiTFSI). The black curve (experiment data) was fitted as the sum of two exponential functions, assigned to the reduction of the residual Li_2S_8 and Li_2S_6 (blue and red, respectively), and a peak from the Li_2S electrodeposition. (b) The extracted current response of Li_2S electrodeposition (black) was compared to the four different nucleation and growth mathematical model using I/I_m vs. t/t_m plot.

Response: We appreciated the reviewer's comments for our manuscript. As the reviewer mentioned, it was reported that lithium ion (Li^+) diffusion coefficient could affect the nucleation morphology of Li_2S (Nature Energy 2017, 2, 813). However, unlike the previous work, which dealt with the Li^+ diffusivity's effect on the formation of submicron-sized Li_2S nucleation, the scale of our Li_2S deposition in Fig. 2 exceeds tens of micrometers. Therefore, due to the low electronic conductivity of Li_2S precipitates, electron transfer through Li_2S surface is no longer available. In the high-DN electrolytes, we suggest completely different mechanism of Li_2S precipitation, in which the reduction to Li_2S and deposition of the product can be separately considered. In this sense, Li_2S binding energy calculation can offer the guidance whether the freely moving Li_2S molecules (through the electrolyte phase) deposit rather on the bulk Li_2S particles than on the carbon electrode surface, offering the detailed process of 3D Li_2S growth enabled by the high-DN anions.

However, it is possible that the initial nucleation process, dictated by Li^+ diffusivity, can influence on or determine the growth morphology of Li_2S . As the reviewer suggested, the Li^+ diffusivity in the three electrolytes were investigated. Fig. R7 displays the Cyclic Voltammetry (CV) curves of the electrolytes collected at different scan rates (0.05 to 0.25 mV s^{-1}). The second cathodic (C2) and the first anodic (A1) current peaks, relevant to the reduction and oxidation of Li_2S , show a linear relationship with the square root values of scan rates. Then, the slopes of the curves represent the Li^+ diffusion rate where other environments are equal (PNAS 2017, 114(5), 840). Since larger numbers of solvent molecules coordinate with Li^+ under the low-DN anion condition (Fig. 6), the LiTFSI electrolyte exhibits the smallest slope and the lowest Li^+ diffusivity. However, the diffusivity of Li^+ is higher for the LiTf electrolyte than the LiBr electrolyte, which does not match with the 3D Li_2S growth trend observed in our manuscript. This diffusivity decrease with the highest-DN anion electrolyte may be due to the extremely high ionic strength of Br^- anion, which triggers an agglomerate form of the Li^+ solvation cluster as observed from the molecular dynamics simulation in Fig. R8.

We again verify whether the diffusion process controls the Li_2S deposition by plotting the current response of the Chronoamperometry (CA) analysis with theoretical models. As we remarked in our manuscript, the CA technique enables the characterization of electrodeposition process. We employed two exponential functions to fit the background and extracted the current response from the electrodeposition of Li_2S ; following the methodology reported in the previous work (Adv. Mater. 2015, 27, 5203). The current vs. time curve was then plotted with the four different 2D nucleation and growth models: Equation R1 to R4 (Faraday Soc. 1962, 58, 2200 / Electrochimica Acta. 1983, 28(7), 879).

$$\text{Kinetic-limited/Instantaneous: } \frac{I}{I_m} = \left(\frac{t}{t_m}\right) \exp\left[-\frac{1}{2}\left(\frac{t^2-t_m^2}{t_m^2}\right)\right] \quad (\text{R1})$$

$$\text{Kinetic-limited/Progressive: } \frac{I}{I_m} = \left(\frac{t}{t_m}\right)^2 \exp\left[-\frac{2}{3}\left(\frac{t^3-t_m^3}{t_m^3}\right)\right] \quad (\text{R2})$$

$$\text{Diffusion-limited/Instantaneous: } \left(\frac{I}{I_m}\right)^2 = \frac{1.9542}{t/t_m} \left[1 - \exp\left(-1.2564 \frac{t}{t_m}\right)\right]^2 \quad (\text{R3})$$

$$\text{Diffusion-limited/Progressive: } \left(\frac{I}{I_m}\right)^2 = \frac{1.2254}{t/t_m} \left[1 - \exp\left(-2.3367 \frac{t^2}{t_m^2}\right)\right]^2 \quad (\text{R4})$$

(where I_m is the peak current, t_m is the corresponding time at which the I_m occurs.)

As shown in the Fig. R9, the current response best fits to the kinetic-controlled/instantaneous nucleation and growth model, indicating that not the diffusion rate of Li^+ but the reaction kinetics of Li_2S formation dominates the overall deposition process of Li_2S . This is in accord with the previous reports in the field of Li_2S nucleation and growth (Nano Lett. 2016, 16, 549 / J. Electrochem. Soc. 2017, 164(4), A917). The discussion to verify the process of 3D Li_2S growth is highly significant, thus all data newly acquired were added in our manuscript as follows.

Revised manuscripts:

(Line 373-381)

In addition, Pan and co-workers recently suggested that 3D Li_2S nucleation and growth were favorably induced with increasing Li^+ diffusion coefficient. Li^+ diffusivity affects the morphology of initial nucleation of Li_2S the scale of which does not hamper the electron transfer through the Li_2S nuclei. However, 3D Li_2S growth up to micron-scale ($\sim 30 \mu\text{m}$) that we examined from the high-DN electrolytes is in a different regime. Since the electrochemical reaction is less likely to happen on the surface of the large-sized Li_2S depositions owing to its extremely low conductivity ($\sim 10^{-13} \text{ S cm}^{-1}$)⁶⁶, the influence of Li^+ diffusion on the particulate Li_2S growth would be less significant.

(Supplementary Fig. 10)

Supplementary Fig. 10 | Fitting of current response to theoretical 2D nucleation and growth models. (a) Background fitting of current vs. time curve for Chronoamperometry (CA) test at 2.00 V (LiTFSI). The black curve (experiment data) was fitted as the sum of two exponential functions, assigned to the reduction of the residual Li_2S_8 and Li_2S_6 (blue and red, respectively), and a peak from the Li_2S electrodeposition³. (b) The extracted current response of Li_2S electrodeposition (black) was compared to the four different nucleation and growth mathematical models^{4,5}.

$$\text{Kinetic – limited/Instantaneous: } \frac{I}{I_m} = \left(\frac{t}{t_m}\right) \exp\left[-\frac{1}{2}\left(\frac{t^2 - t_m^2}{t_m^2}\right)\right] \quad (\text{S1})$$

$$\text{Kinetic – limited/Progressive: } \frac{I}{I_m} = \left(\frac{t}{t_m}\right)^2 \exp\left[-\frac{2}{3}\left(\frac{t^3 - t_m^3}{t_m^3}\right)\right] \quad (\text{S2})$$

$$\text{Diffusion – limited/Instant: } \left(\frac{I}{I_m}\right)^2 = \frac{1.9542}{t/t_m} \left[1 - \exp\left(-1.2564 \frac{t}{t_m}\right)\right]^2 \quad (\text{S3})$$

$$\text{Diffusion – limited/Progress: } \left(\frac{I}{I_m}\right)^2 = \frac{1.2254}{t/t_m} \left[1 - \exp\left(-2.3367 \frac{t^2}{t_m^2}\right)\right]^2 \quad (\text{S4})$$

(where I_m is the peak current, t_m is the corresponding time at which the I_m occurs.)

The current response fits to the kinetic-controlled/instantaneous nucleation and growth model based on I/I_m vs. t/t_m plots.

5. Although the electrochemical measurements are quite comprehensive and solid, tests with more cycle numbers and under higher current densities are required to examine the long-term serviceability and rate capability. Besides, the potential of these electrolytes with high-DN anions for practical Li-S batteries operated at a lean-electrolyte condition is suggested to be commented on.

Fig. R10 | (a) Charge and discharge profiles at the first cycle (0.5 C), (b) comparison of the charge (closed circle) and discharge (open circle) capacities during the cycling at 0.5 C, (c) coulombic efficiencies for 80 charge/discharge cycles at 0.5 C, and (d) rate capability test with increasing current density from 0.1 C to 1 C and recovering to 0.1 C. The electrolytes consist of 0.2 M LiPS (Li₂S₈ based) with 1 M Li salts (LiX, X= TFSI,

Tf, or Br) / 0.2 M LiNO₃ / DOL:DME (1:1)

Fig. R11 | Lithium polysulfide (LiPS, Li₂S₈ based) solubility changes of 1 M LiTFSI DOL:DME(1:1) electrolyte with decreasing the electrolyte to sulfur ratio (E/S ratio) of 20:1, 5:1, and 3:1.

Response: We are grateful for what the reviewer comments. First, as suggested, the electrochemical cell tests with a higher current density were conducted using the three different salt anions. The theoretical areal capacities of the cells were set to 1.68 mAh cm⁻². As shown in Fig. R10, the LiTFSI electrolyte with the lowest-DN delivered a low capacity, whereas the LiTf and LiBr electrolytes realized high discharge capacities of 994 and 1310 mAh g⁻¹, respectively. Due to the higher overvoltage and faster electrolyte consumption under the higher current density, its specific capacity and cycle life were reduced. Nevertheless, the high-DN anions can retard the surface passivation and enable the higher sulfur utilization even at the high current density. These cycling data under the high current density were added to the revised manuscript.

Regarding the lean-electrolyte condition, we agree with the reviewer's proposal to test the effectiveness of the high-DN anion approach at low electrolyte to sulfur (E/S) ratio. Unfortunately, the catholyte-based Li-S system, which we employed to examine the electrochemical effect and the deposition morphology of 3D Li₂S growth, is not suitable to fabricate the low electrolyte/sulfur (E/S) ratio Li-S batteries. It is mainly due to the solubility limit of LiPS species in ether-based electrolytes (J. Electrochem. Soc. 2017, 164(4), A917). Compared to the catholyte sample with the E/S ratio of 20:1, low electrolyte containing catholytes do not completely dissolve the lithium polysulfides (LiPSs) as shown in Fig. R3, thus cannot be used for Li-S cells. We have acknowledged the inadequacy of the catholyte based system; therefore, we are currently preparing for the follow-up study to investigate the effectiveness of high-DN anions under low E/S ratio conditions, using a different type of Li-S batteries.

Revised manuscript:

(Line 122-125)

Moreover, even at a higher current density of 0.5 C, the electrolytes with high-DN salt anions maintained their role in enhancing the discharge capacities (LiTf and LiBr, 994 and 1310 mAh g⁻¹, respectively) and enabled reasonably stable cycling (Supplementary Fig. 3).

(Supplementary Fig. 3)

Supplementary Fig. 3 | Electrochemical performances at higher current densities with the LiTFSI (red), LiTf (blue), and LiBr (green) electrolytes (a) Charge and discharge profiles at the first cycle (0.5 C) (b) comparison of the charge (closed circle) and discharge (open circle) capacities during the cycling at 0.5 C, (c) coulombic efficiencies for 50 charge/discharge cycles at 0.5 C, and (d) rate capability test with increasing current density from 0.1 C to 1 C and recovering to 0.1 C. The electrolytes consist of 0.2 M LiPS (Li_2S_8 based) with 1 M Li salts (LiX , $\text{X} = \text{TFSI}$, Tf , or Br) / 0.2 M LiNO_3 / DOL:DME (1:1)

Reviewers' Comments:

Reviewer #2:

Remarks to the Author:

The manuscript has been perfectly revised with very comprehensive data support. It should be published on Nature Communication as soon as possible. I believe that it will bring very fresh insights to Li-S battery research and the whole electrochemistry community.